# Movement initiation and grasp representation in premotor and primary motor cortex mirror neurons

**Steven Jack Jerjian[1†], Maneesh Sahani[2], Alexander Kraskov[1]\***

[1]Department of Clinical and Movement Neurosciences, UCL Institute of Neurology, London, United Kingdom; [2]Gatsby Computational Neuroscience Unit, University College London, London, United Kingdom

**Abstract** Pyramidal tract neurons (PTNs) within macaque rostral ventral premotor cortex (F5) and (M1) provide direct input to spinal circuitry and are critical for skilled movement control. Contrary to initial hypotheses, they can also be active during action observation, in the absence of any movement. A population-level understanding of this phenomenon is currently lacking. We recorded from single neurons, including identified PTNs, in (M1) (n = 187), and F5 (n = 115) as two adult male macaques executed, observed, or withheld (NoGo) reach-to-grasp actions. F5 maintained a similar representation of grasping actions during both execution and observation. In contrast, although many individual M1 neurons were active during observation, M1 population activity was distinct from execution, and more closely aligned to NoGo activity, suggesting this activity contributes to withholding of self-movement. M1 and its outputs may dissociate initiation of movement from representation of grasp in order to flexibly guide behaviour.

**\*For correspondence:**
a.kraskov@ucl.ac.uk

**Present address:** [†]Zanvyl Krieger Mind/Brain Institute, Johns Hopkins University, Baltimore, United States

**Competing interests:** The authors declare that no competing interests exist.

## Introduction

The defining property of mirror neurons (MNs) is that they modulate their firing both when a monkey performs an action, and when it observes a similar action performed by another individual (*Gallese et al., 1996*; *Rizzolatti and Fogassi, 2014*). Since their discovery in the macaque rostral ventral premotor cortex (F5), cells with mirror-like properties have been identified in parietal areas (*Fogassi et al., 2005*; *Bonini et al., 2010*; *Lanzilotto et al., 2019*), dorsal premotor cortex (PMd) (*Cisek and Kalaska, 2004*; *Papadourakis and Raos, 2019*), and even M1 (*Tkach et al., 2007*; *Dushanova and Donoghue, 2010*; *Vigneswaran et al., 2013*). MNs thus appear to be embedded within a parieto-frontal network (*Bonini, 2017*; *Bruni et al., 2018*) integral to the execution of visually-guided grasp (*Jeannerod et al., 1995*; *Borra et al., 2017*). The widespread activity within this circuitry during action observation takes place in the absence of detectable movement or muscle activity, despite the finding that even PTNs, which project directly to the spinal cord, can exhibit mirror properties (*Kraskov et al., 2009*; *Vigneswaran et al., 2013*).

F5 MNs often show similar levels of activity during execution and observation (*Gallese et al., 1996*; *Kraskov et al., 2009*), however in M1- there is typically a reduced level of firing during observation relative to execution (*Vigneswaran et al., 2013*; *Kraskov et al., 2014*). By design, most action observation paradigms require movement suppression, and the disfacilitation of spinal outputs therefore provides a rational, threshold-based explanation for why movement is not produced. However, there is substantial empirical evidence of both facilitation and suppression during movement execution in PTNs (*Kraskov et al., 2009*; *Quallo et al., 2012*; *Vigneswaran et al., 2013*; *Soteropoulos, 2018*), which suggests a more nuanced relationship between PTN activity and movement. At the spinal level, PTNs not only excite motoneurons via cortico-motoneuronal (CM) projections (*Porter and Lemon, 1993*; *Rathelot and Strick, 2006*), but also exert indirect effects via

segmental interneuron pathways, which in turn display their own complex activity before and during movement (*Prut and Fetz, 1999*; *Takei and Seki, 2013*). A dynamical systems approach (*Shenoy et al., 2013*) has recently suggested that movement-related activity unfolds in largely orthogonal dimensions to activity during action preparation, such that movement is implicitly gated during movement preparation (*Kaufman et al., 2014*; *Elsayed et al., 2016*), and a similar mechanism has been hypothesised to operate during action observation (*Mazurek et al., 2018*) and action suppression (*Pani et al., 2019*). While the roles of F5 and M1 during the execution of visually-guided grasp have been studied extensively (*Umilta et al., 2007*; *Davare et al., 2008*; *Schaffelhofer and Scherberger, 2016*), a more systematic understanding of the differences between action execution and observation activity in these two key nodes in the grasping circuitry could provide important insights into dissociations between representation of potential actions at the cortical level, and recruitment of descending pathways and muscles for actual action execution (*Schieber, 2011*). Along these lines, recent work comparing MNs in premotor and motor cortex found premotor MNs, but not those in M1, showed similar state transitions in execution and observation (*Mazurek et al., 2018*). State-space analyses have also previously found that F5 and the upstream anterior intraparietal area (AIP) exhibit different dynamics during immediate and delayed grasping actions (*Michaels et al., 2018*).

Although disfacilitation of selected spinal outputs in M1 during action observation was suggestive of a mechanism to avoid unwanted self-movement (*Vigneswaran et al., 2013*), it is unclear how this fits with recent evidence indicating that movement generation is mediated by patterns of covariation at the population level (*Churchland et al., 2012*; *Kaufman et al., 2014*), rather than a ramping-to-threshold mechanism. Furthermore, if aspects of observation activity reflect a true neural correlate of movement suppression, an observable relationship with other forms of movement suppression might be expected. While previous work has examined grasp representation in F5 during inaction conditions (*Bonini et al., 2014b*), and reported little overlap between MNs and neurons encoding self-action withholding, this has not been examined in M1. Interleaved action and inaction within peri-personal space may also provide a more ethologically valid framework for investigating movement suppression during action observation. Here, we sought to explore these two issues by comparing the activity of MNs in M1 and F5 of two macaque monkeys, while they switched between executing, observing, and withholding reach-to-grasp and hold movements on a trial-by-trial basis. Electrical stimulation in the medullary pyramid was used to antidromically identify PTNs, and we leveraged the precise timing of task events within a naturalistic experimental paradigm to assess and compare the patterns of discharge of different populations of neurons across task conditions. We first investigated the relationship between execution and observation population activity among F5 and M1 MNs. We then examined whether neural trajectories which diverged from the movement subspace during action observation occupied a putative active 'withholding' subspace, by comparing observation activity to activity when monkeys were simply cued to withhold their own actions.

## Results

We recorded single neurons in F5 and M1 of rhesus macaques performing and observing reach-to-grasp and hold actions (*Figure 1*), and investigated the population-level differences in execution and observation activity which could explain how overt movement is withheld during the latter condition. We then considered whether observation activity contained more general signatures of movement suppression by comparing modulation during the action observation condition, where monkeys were required to remain still, to neural activity when monkeys were explicitly cued to withhold their own movement.

### EMG activity and behaviour during task performance

Monkeys were trained to a high level of performance before recording (>90% correct trials per session). For both monkeys, reaction and movement times were significantly faster than human experimenters (*Table 1*, Wilcoxon sign-rank test on session averages, all $p < 1 \times 10^{-13}$). As the trapezoid object was positioned contralateral to the reaching (right) arm, monkey movement times were 30–50 ms longer than those for the sphere (Wilcoxon sign-rank test, both monkeys $p < 1 \times 10^{-7}$). To verify that neural activity during action observation and withholding was not confounded by muscle activity, we simultaneously recorded electromyography (EMG) from up to 12 hand and arm muscles.

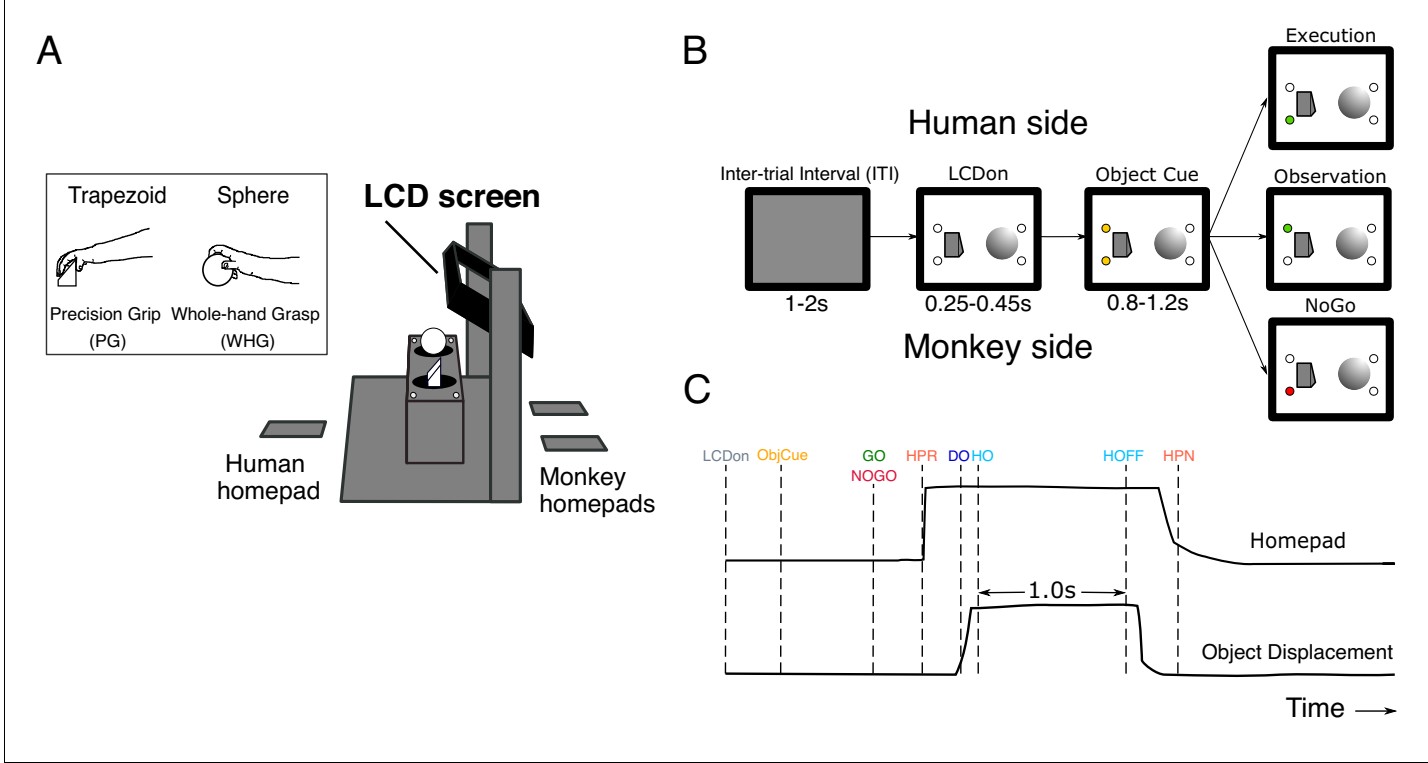

**Figure 1.** Experimental task design. (A) Schematic of the custom-built experimental box, showing target objects, their corresponding LEDs, LCD screen, and homepads. Inset shows the trapezoid and sphere objects, and the respective precision and whole-hand grasps performed by the monkeys on execution trials. (B) Pseudo-random trial presentation sequence, shown as 2-D schematic. All trials began in the same way, with the object area illuminated (LCDon), and upcoming object/grasp cued (e.g. trapezoid, precision grip [PG]). Each trial was then indicated as Execution (green LED on monkey side), Observation (green LED on human experimenter side), or NoGo (red LED on monkey side). (C) Homepad and object displacement signals on Go trials, and digital task events. LCDon LCD screen becomes transparent, ObjCue, object cue (amber LED); Go/NoGo, green/red LED; HPR, homepad release; DO, displacement onset; HO, hold onset; HOFF, hold offset; HPN, homepad return.

During action execution, we observed characteristic patterns of EMG for each grasp (*Figure 2A*). In the action observation and NoGo conditions, on the other hand, EMG activity was negligible (*Figure 2—figure supplement 1*, observation and NoGo are plotted at x10 gain). We further quantified and compared the relative magnitude of EMG during the Baseline (LCDon-ObjCue) and Reaction period (Go-HPR for execution, 0–300 ms after the imperative cue for observation and NoGo) across conditions and sessions (*Figure 2B,C*; see Materials and methods). Across recordings, the magnitude of EMG during Observation and NoGo Reaction periods were not significantly different from baseline ($t_{1,92} = 0.008$, p=0.99, and $t_{1,92}$ = -0.55, p=0.58, respectively), suggesting that the trained monkeys were able to appropriately withhold activity in the passive conditions. Both conditions were very different from Execution Reaction (observation: $t_{1,92}$ = 11.64, NoGo: $t_{1,92}$ = 11.55, both p<0.00001), consistent with onset of EMG activity in the lead-up to monkey homepad release (HPR). Nevertheless, to fully exclude the possibility that individual trials with subtle EMG activity could contaminate observation and NoGo neural responses, we employed an iterative procedure to exclude passive trials with detected EMG activity (see Materials and methods).

## Effects of repetitive intracortical microstimulation

We delivered repetitive intra-cortical microstimulation (rICMS) at 57 sites containing M1-PTNs, 124 sites with unidentified neurons (UIDs) in M1, and 111 sites in F5. Finger or thumb effects were elicited at 27/57 M1-PTN sites, 89/124 M1-UID sites, and 75/111 F5 sites. The majority of these sites had low thresholds in M1 (20/27 (74.1%) and 76/89 (85.4%) ≤20μA, PTNs and UIDs respectively), but not in F5 (27/75 (36.0%)).

**Table 1.** Behaviour during recording sessions for basic mirror task.

RT, reaction time; MT, movement time. Reaction time was defined as the time between the Go cue and homepad release (HPR), and movement tie as the time between HPR and object displacement onset (displacement onset [DO]). Values denote mean ± SEM of median values from each session, rounded to nearest millisecond.

| | M48 | | | | M49 | | | |
| --- | --- | --- | --- | --- | --- | --- | --- | --- |
| | Monkey | | Human | | Monkey | | Human | |
| | PG | WHG | PG | WHG | PG | WHG | PG | WHG |
| RT (ms) | 310 ± 25 | 267 ± 22 | 469 ± 38 | 442 ± 44 | 272 ± 22 | 268 ± 16 | 412 ± 48 | 401 ± 41 |
| MT (ms) | 306 ± 20 | 279 ± 14 | 430 ± 31 | 374 ± 38 | 404 ± 23 | 351 ± 20 | 520 ± 39 | 532 ± 45 |

## Database

Single neurons were recorded across 25 sessions in M48, and 40 sessions in M49 (in 93 separate recordings). After discarding EMG-contaminated observation and NoGo trials, we were left with a total of 302 neurons recorded for at least 10 trials per grasp for both execution and observation conditions (*Table 2*), on which 296 were also recorded for at least 7 NoGo trials per grasp. 187 units were recorded in M1, and 115 in F5. 59 M1 neurons were identified as PTNs; the remaining 128 were UIDs. F5-PTNs were recorded (15 in M48, 8 in M49), however the total number of MNs was relatively low (15), rendering it difficult to extract meaningful conclusions within this population alone. Given the weak contribution of F5 PTNs to descending control of grasp (*Dum and Strick, 1991*; *He et al., 1993*; *Cerri et al., 2003*; *Shimazu et al., 2004*), we elected to consider all F5 neurons (23 PTNs and 92 UIDs) as one population. *Figure 3* shows an MRI rendering of all penetrations in both subjects in which single units were recorded, confirming that the majority of recordings were made near the hand area of M1, and posterior to the inferior limb of the arcuate sulcus.

## Single-neuron responses during execution and observation

The complex naturalistic task set-up evoked a wide variety of responses in recorded neurons, particularly during action execution, and a substantial proportion of neurons also showed responses during action observation. *Figure 3* shows three M1-PTNs and one F5-UID, which all showed time-dependent modulation during execution and observation, with varying levels of similarity between the responses in the two conditions. The two M1-PTNs in (A) and (B) showed dynamic changes in activity during the reaching and grasping period, with smaller and steadier increases in activity from baseline during observation (bottom panels). The third M1-PTN (*Figure 4C*) completely silenced during both execution and observation hold, before showing some rebound at the end of this period. The F5-UID in *Figure 4D* transiently and dramatically increased firing during both execution and observation around the time of grasp for both objects, and maintained a steady, lower level of firing during execution, but not observation hold.

For each neuron, we first assessed the statistical significance of changes in firing rate separately during execution and observation across baseline and two task epochs (Reach and Grasp/Hold) via 2-way ANOVA (see Materials and methods). During execution, 278/302 neurons (92.1%) showed a main effect of epoch, and 216 (71.5%) had an epoch × grasp interaction effect. During observation, 204/302 (67.6%) showed a main effect of epoch, and 59 (19.5%) showed an interaction effect. The proportion of interaction effects was significantly higher during execution than observation (chi-squared test, $\chi^2_{1,302}$ = 164.6, p<0.00001), consistent with more frequent grasp specificity during action execution. Based on results from the 2-way ANOVA and post-hoc comparisons to baseline (see Materials and methods), 282/302 (93.4%) neurons were considered modulated during execution, and 174 (57.6%) during observation. 169 neurons (56.0% of total) were considered as MNs based on significant modulation during both execution and observation.

## Population-level activity during execution and observation

The extent of modulation during action observation may differ across premotor and motor cortex at the population level, and given the relative contributions of these two areas to the CST, these differences are likely to have important implications for the potential effects of observation activity on

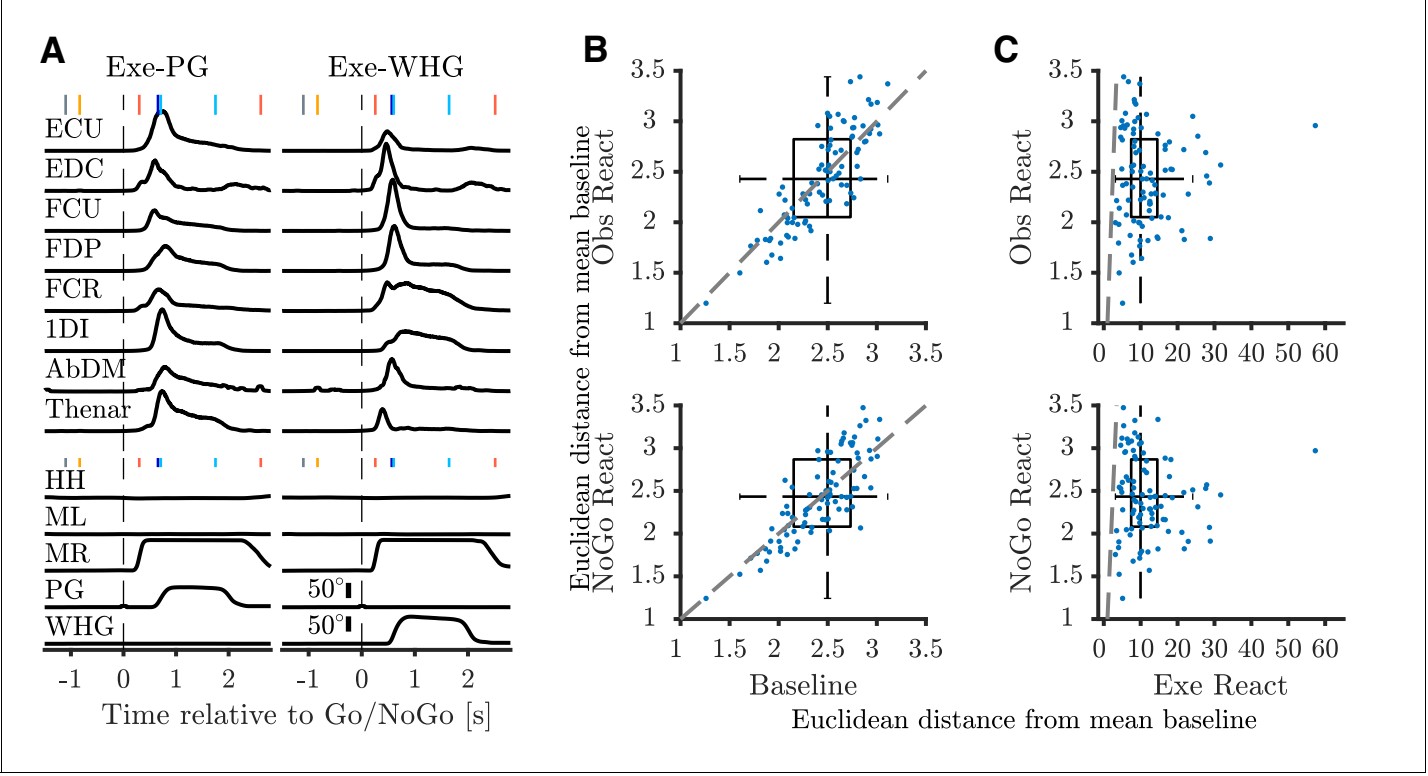

**Figure 2.** EMG during task. (**A**) Average execution EMG traces during a single session in M48. Top panels show pre-processed, rectified, and normalized EMG activity for different muscles with clean recordings for precision grip (PG) (left), and whole-hand grasp (WHG) (right). Bottom panels show corresponding average homepad and object displacement signals. Vertical markers at top of each trace indicate median time of task events relative to Go/NoGo cue (vertical dashed lines); colour coded as in *Figure 1C*. ECU, extensor carpi ulnaris; EDC, extensor digitorum communis; FCU, flexor carpi ulnaris; FDP, flexor digitorum profundus; FCR, flexor carpi radialis; 1DI, first dorsal interosseous; AbDM, abductor digiti minimi; HH, human homepad; ML, monkey left homepad; MR, monkey right homepad; PG precision grip; WHG, whole-hand grasp. (**B**) 2-D boxplot representation of Euclidean distance across muscles from mean baseline EMG. Blue dots show median value for each session (n = 93 total), dashed grey line denotes unity. (**C**) Distance from mean baseline of Observation React (top) and NoGo Reaction (bottom) periods vs. Execution Reaction.

The online version of this article includes the following figure supplement(s) for figure 2:

**Figure supplement 1.** Example EMG traces during all conditions.

downstream targets. The heatmaps in *Figure 5A–C* show the time-resolved net normalized firing rate during precision grip (PG) execution and observation across the three MN sub-populations, and histograms show the averages during execution and observation for the PG facilitation-facilitation and facilitation-suppression units (for whole-hand grasp (WHG), see *Figure 5—figure supplement 1*). Within each sub-population, we found both facilitation and suppression responses relative to baseline during execution and observation, and the relationship between activity in the two conditions was variable. For the commonest group of identified MNs, net normalized activity of facilitation-facilitation (F-F) MNs (those which increased their activity during execution and observation) was generally larger during execution movement than observation, particularly in M1-PTNs (*Figure 5A*, top right panel). Net execution activity in the F-F population showed a 3.2 to 4.1-fold (PG and WHG, respectively) increase from observation activity at the moment of grasp (DO). The average across the two grasps was a 3.5-fold increase (average net normalized activity in execution: 0.482, observation: 0.136), and the same ratios in M1-UIDs and F5 F-F populations were 2.32 and 1.52, respectively, revealing a progressive decline in the amplitude difference between execution and observation through the three sub-populations. Notably, although the overall magnitude of execution and observation activity in the F-F M1-PTN population were relatively similar at the time of movement onset (HPR), the trajectories of the neural activity around this time were markedly different (*Figure 5A* and *Figure 5—figure supplement 1A*, top right panels), with a brief rise and fall during execution before the eventual large increase in activity, and a gradual, later increase during

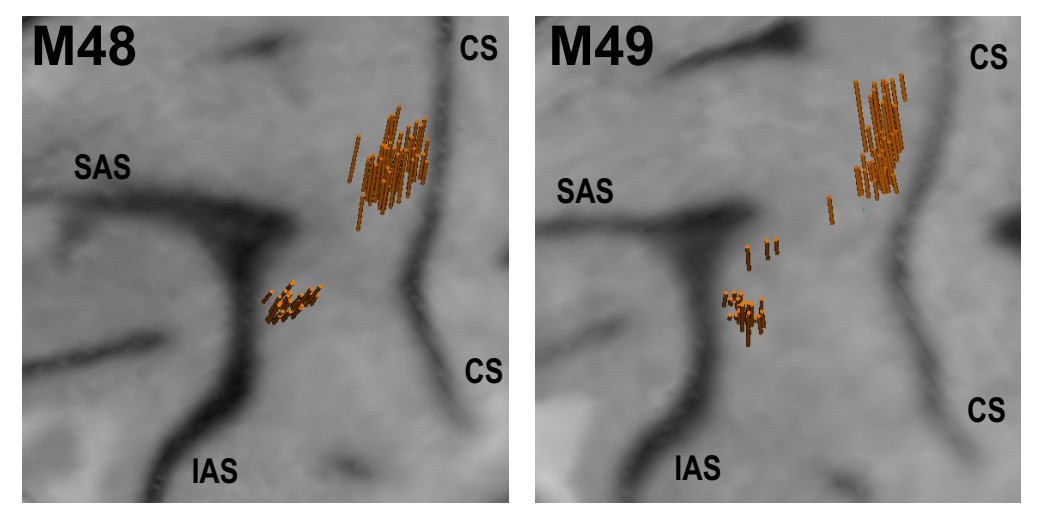

**Figure 3.** Structural MRI showing angle and location of electrode penetrations in which single-units were recorded in left F5 and M1 of M48 (left panel), and M49 (right panel). The brain surface was estimated in the BrainsightVet software (Rogue Research Inc) using a curvilinear approximation method. Penetration locations and orientations were estimated via a geometrical transformation between recording drive and MRI coordinates. CS - central sulcus, SAS - superior limb of arcuate sulcus, IAS - inferior limb of arcuate sulcus.

observation. Thus, while the amplitude differences seen during execution and observation grasp in the F-F populations align with the ongoing behaviour (movement or no movement), we considered whether divergences in the temporal pattern of activity in different sub-populations after the Go cue could provide a clearer insight into the differences contributing to movement generation or suppression in the two conditions.

To compare the time-varying pattern of activity during action execution and observation, we first computed the correlation between execution and observation activity across each MN subpopulation during different task epochs (*Figure 6* and *Figure 6—figure supplement 1*). During ObjCue, when trials were identical from the monkey's perspective, all populations showed a strong, significant correlation between the two conditions (r > 0.9, p<1 $\times$ 10$^{-32}$, *Figure 6* left inset, and *Figure 6—figure supplement 1A*). Contrastingly, activity patterns during the early stages of the reach were markedly different (*Figure 6—figure supplement 1*, middle row). This was particularly the case in M1-PTNs, which showed no significant relationship between execution and observation activity at this stage of the task (r = 0.15, p=0.2, *Figure 6A*, middle inset). M1-UIDs and F5 populations were also less well correlated during this period than before the Go cue, although the correlations remained significant (p<1 $\times$ 10$^{-5}$). During the Hold period, execution and observation were again significantly correlated (p $\leq$ 1e-10, *Figure 6—figure supplement 1C*). We also compared the observed correlation values to null distributions created by shuffling the observation vector so that within-unit relationships were lost (*Figure 6*). Correlations during the early reach period were significantly greater than all values in the null distribution for M1-UIDs and F5 (both p=0.001, permutation test), but not M1-PTNs (p=0.15), confirming that the relationship between execution and observation at the population level was particularly weak in M1-PTNs during the early reaching period.

To assess the temporal stability of cross-condition similarity, we performed a cross-temporal pattern analysis using time-resolved PSTHs, by computing the correlation between net normalized activity at each timepoint with that of every other timepoint (*Figure 6B*). The diagonal of this matrix therefore roughly corresponds to the epoch-based correlation values above. Activity prior to the Go cue, and during the hold period, was generally well correlated across the two conditions in all three populations. F5 neurons showed stronger correlations between the object cue and later hold periods, which was not apparent for M1-PTNs, indicating that the pattern of activity in these two periods was more consistent in F5.

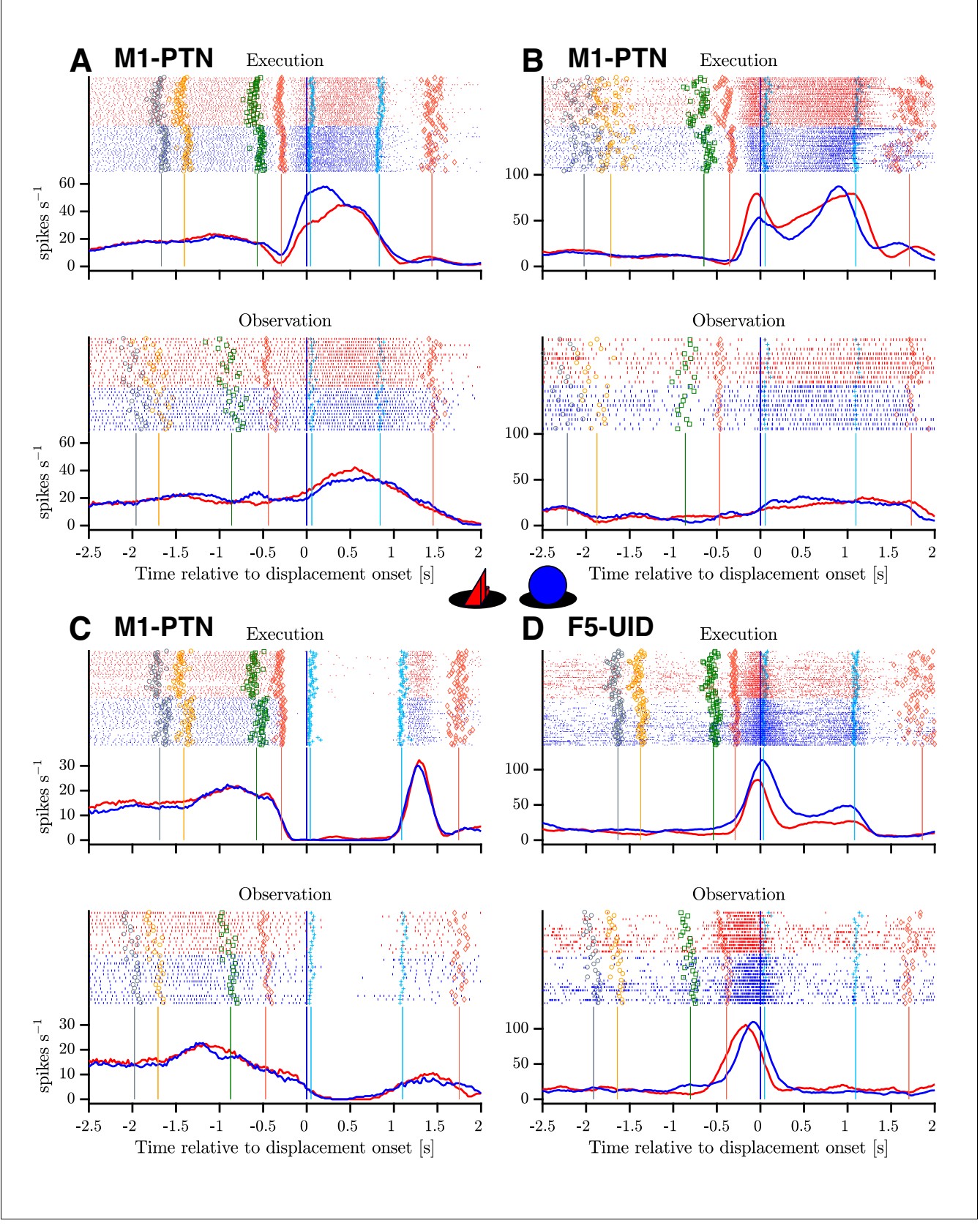

**Figure 4.** Example mirror neurons in M1 and F5. Raster and histogram representations of single neuron activity during execution (top panels) and observation (bottom panels). (**A–C**) Three M1-PTNs, showing varying relationships between execution and observation activity. (**D**) F5-UID showing substantial modulation during both conditions. Units in (**A**), (**C**) and (**D**) were recorded in M48, (**B**) was recorded in M49. Activity is aligned to object displacement (DO). Rasters are split by grasp (PG and WHG, objects shown in central inset) and condition for visualization purposes, although trials were presented in a pseudo-randomised order during recording. Single trial events are indicated on raster plots (LCDon, Object Cue, Go, HPR, HO, HOFF, HPN), and median times relative to alignment are shown on histograms. Event colours are as shown previously (*Figure 1C*): LCDon - grey; Object Cue - orange; Go - green; HPR and HPN - magenta; HO and HOFF - cyan). For histograms, firing rates were calculated in 20 ms bins and boxcar-smoothed (200 ms moving average).

We next used PCA to examine the nature of time-varying patterns of activity across action execution and observation in each sub-population within a movement subspace. PCA identifies the dominant modes, or dimensions of neural activity within the full dimensional space, which capture the majority of the variance in the data. The activity of the same neurons recorded during a different behaviour or time period can then be compared to the first based on the similarity of the covariance across neurons, which will result in similar or different projections upon the defined dimensions. This holds advantages over unweighted averaging of neural activity in different conditions, which also reduces dimensionality, but altogether sacrifices information regarding the relationships between different neurons and conditions. We defined a movement subspace empirically for each sub-population, using trial-averaged activity during execution reach and grasp, and then visualized evolution of execution (green) and observation (purple) trajectories across the first 2 axes of this execution movement subspace (*Figure 7A*). PG activity prior to the Go Cue was similar and overlapping for the two conditions and showed little variance in the movement subspace, reflected by the minimal evolution of the trajectories until this point. After the Go cue in execution, activity in each population then progressively evolved through different stages of the trial through HPR and DO, as indicated by the arrows, spanning the movement subspace for each grasp (PG: *Figure 7A* and WHG: *Figure 7—figure supplement 1A*). During action observation, M1-PTNs (*Figure 7A*, left) and M1-UIDs (*Figure 7A*, middle) showed a highly collapsed trajectory, suggesting little similarity between population activity in execution and observation after the Go cue. F5 population activity, on the other hand, followed a qualitatively similar, albeit smaller trajectory to that seen during execution, with ordered progression through stages of the task (*Figure 7A*, right). For each population, we quantified the level of variance captured on these axes for both execution and observation. While the PCA method ensured that three dimensions captured the majority of the variance (>90%) of the execution data for all three populations (*Figure 7B–D* and *Figure 7—figure supplement 1B–D*, left panels), captured observation variance was relatively low for both grasps (¡20% in all cases). The ratio of this variance, to the maximum possible variance which could be captured within the observation data constituted a normalized measure of alignment (*Figure 7B–D* and *Figure 7—figure supplement 1B–D*, right panels, purple lines, see Materials and methods). To quantify the significance of this overlap relative to what could be expected simply by chance, we compared this alignment to a null distribution of alignment of random orthonormal dimensions. During movement, we found that only F5 showed an alignment between observation and execution greater than expected from chance for both grasps (PG p=0.006, WHG: p=0.0007, upper-tailed permutation test). In M1-PTNs and M1-UIDs, on the other hand, alignment was not significantly different to chance (both grasps and populations p>0.05). To assess whether our measures of alignment were sensitive to potential EMG contamination, we repeated subspace analyses by projecting observation PSTHs compiled via a median split of all trials based on EMG magnitude during the Reaction period after the

**Table 2.** Number of single-units recorded in each monkey and sub-population for at least 10 execution and 10 observation trials per grasp (after removal of contaminated trials).

|         | M48 | M49 | Total |
|---------|-----|-----|-------|
| M1-PTN  | 35  | 24  | 59    |
| M1-UID  | 77  | 51  | 128   |
| F5      | 72  | 43  | 115   |
| Total   | 184 | 118 | 302   |

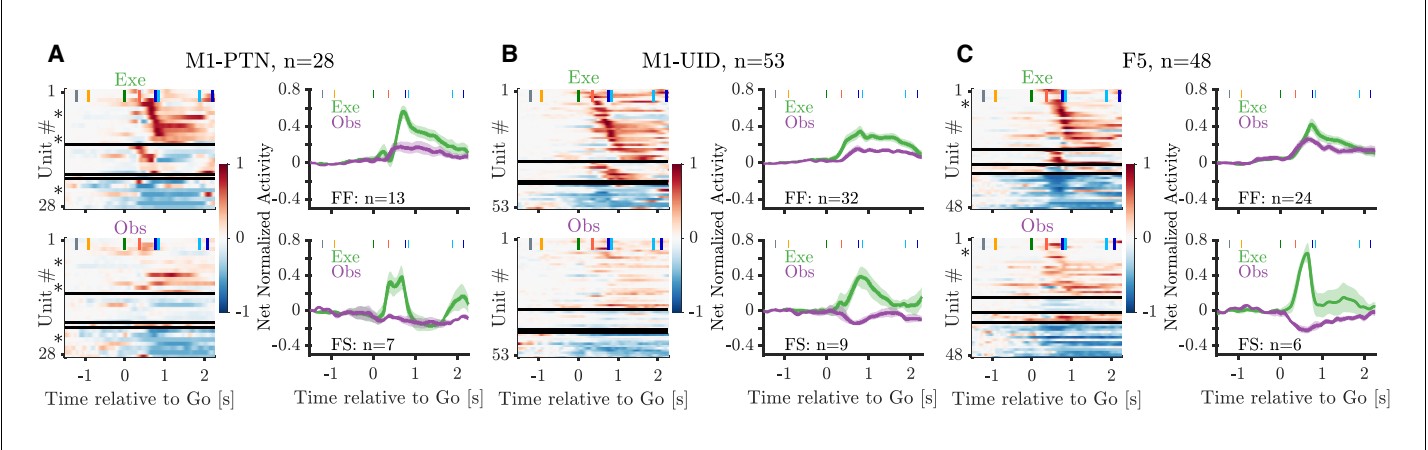

**Figure 5.** Mirror neuron population activity during PG. (**A**) Left panels show heatmaps of net normalized activity of PG MNs within the M1-PTN population. Neurons are split into facilitation-facilitation, facilitation-suppression, suppression-facilitation, and suppression-suppression categories based on the sign of their modulation during action execution (top) and observation (bottom) relative to baseline. Horizontal black lines mark splits between categories. Within each category, neurons are sorted based on the latency of their absolute peak response during execution (peak calculated between GO and HO+0.5 s). Asterisks denote units shown in *Figure 4*. Population averages are shown for F-F (top right panel) and F-S categories (bottom right panel). (**B**) Same as (**A**) but for M1-UIDs. (**C**) Same as (**A**) but for F5.

The online version of this article includes the following figure supplement(s) for figure 5:

**Figure supplement 1.** WHG Heatmaps and population averages.

observation Go cue (without any prior EMG-based exclusion) . We found that PG M1-PTN alignment was weakly significant for the split containing trials with above-median EMG (p=0.048), but not for the split containing trials with the lower EMG level (p=0.15). This was not the case for WHG, nor any M1-UID (all p>0.05), or F5 split (all p<0.05). Although EMG contamination of observation and NoGo trials was small and rare such that overall changes in alignment were modest, these results suggest that, particularly for M1-PTN, small increases in EMG during observation may increase the share of neural activity captured by the movement subspace.

To address whether the relationship between the two grasps in each sub-population was similar or different in execution and observation, we compared bootstrapped alignment values obtained via projection of one grasp's activity onto the subspace defined by the other grasp, for execution and observation separately. Projecting WHG activity onto the PG subspace (*Figure 7E*), we found that alignment values were similar for execution and observation in F5 (mean alignment: 0.44 and 0.57 respectively, p=0.15 via permutation test), but were significantly greater during observation in both M1 populations (M1-PTNs: 0.38, 0.72, p=0.004; M1-UIDs 0.41, 0.69, p=0.008). The same was true when projecting PG activity onto WHG subspaces (*Figure 7F*, M1-PTNs: 0.37, 0.72, p=0.005; M1-UIDs: 0.41, 0.68, p=0.007, F5: 0.48, 0.55, p=0.279). Taken together, these analyses suggest that grasp representation is more similar across execution and observation in F5, whereas in M1 the representation of grasps during execution appears to have little bearing on their representation during observation.

## Movement suppression during action observation

The finding that observation activity, particularly in M1 populations, diverges from execution activity after the Go cue, and resides in a largely separate subspace, is consistent with previous suggestions that disfacilitation of spinal outputs during action observation may provide a mechanism for withholding of self-movement. On its own however, this does not address whether movement is withheld during observation simply via a net 'absence' of execution-like activity, or whether there is structured suppression-related activity during action observation. To explore this latter hypothesis, we considered whether the structure of activity during action observation after the Go cue shared parallels with activity during a simple and well-studied form of movement suppression, when the monkey is explicitly cued to withhold movement via a NoGo cue. *Figure 8A* shows four single neurons recorded during PG execution, observation, and NoGo conditions. The activity patterns of the first

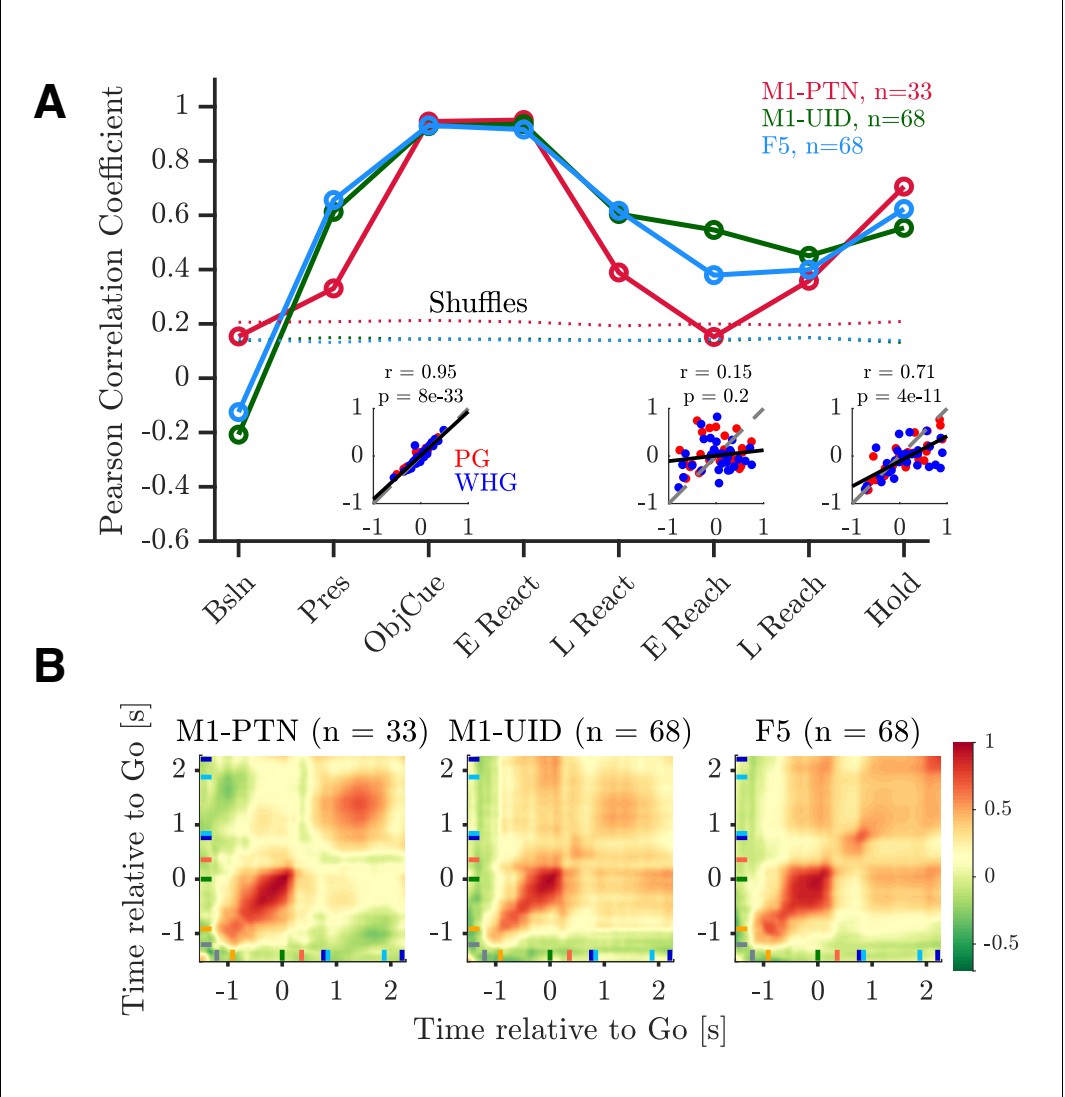

**Figure 6.** Relationship between execution and observation activity. Pearson correlation coefficients of execution and observation activity shown for each epoch and MN sub-population. Dotted lines represent 95th percentiles of null distribution calculated via shuffling neurons. Insets show net normalized activity during execution and observation in M1-PTNs during Object Cue, Early Reach, and Hold epochs. PG and WHG are shown in red and blue, respectively, correlations are calculated across both grasps.

The online version of this article includes the following figure supplement(s) for figure 6:

**Figure supplement 1.** Execution and observation correlation scatter plots.

M1-PTN and M1-UID (left two panels) became clearly different for movement and non-movement around 100–150 ms after the Go/NoGo cue, but showed comparatively little difference between observation and NoGo. By contrast, the activity of the second M1-PTN (middle right panel), which is the same neuron as shown in *Figure 4A*, was clearly different for all three conditions. The F5 neuron (*Figure 8A*, far right) discharged in a similar way for execution and observation, first decreasing then increasing activity, while increasing activity in the NoGo condition. Using all neurons with at least 10 trials recorded per task condition, we trained a maximum correlation coefficient classifier to decode condition (execution-observation-NoGo) for each cortical population (*Figure 8B*). Across all three populations, the decoder was able to distinguish condition with high accuracy from 100 to 150 ms after the Go/NoGo cue was given. We hypothesised that this could be largely driven by very reliable decoding of execution, which often shows greater variation in firing rates, and therefore also trained

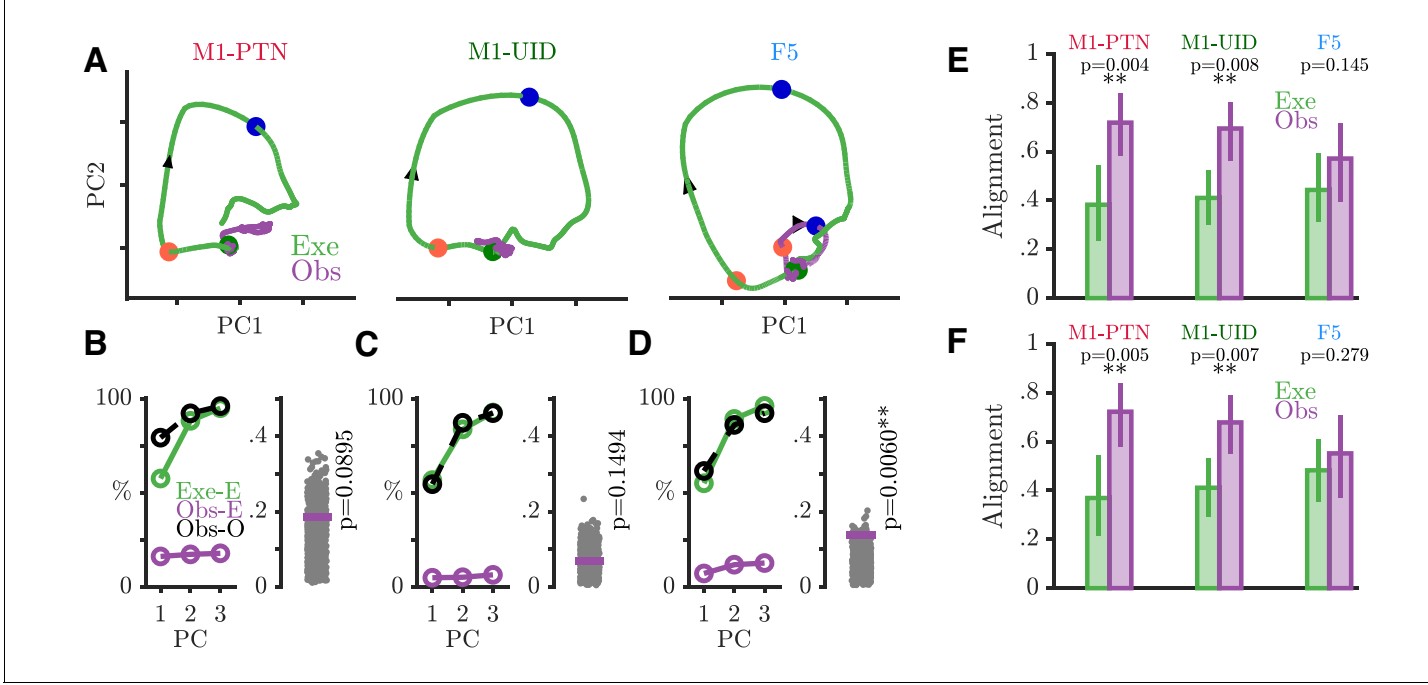

**Figure 7.** Execution and observation activity within a movement subspace. (**A**) Traces showing the evolution of M1-PTN, M1-UID and F5 population activity within a 2-D movement subspace (defined by movement execution activity) across PG execution (green) and observation (purple) trial conditions. Larger coloured circles on each trajectory mark key events (green - Go, orange - HPR, blue - DO) used for multiple alignment of neural activity, and arrows on trajectories indicate direction of time. (**B**) M1-PTNs *Left Panel:* Cumulative variance captured by the first three principal axes. Exe-E (green), execution variance in execution subspace; Obs-E (purple), observation variance in execution subspace; Obs-O (black dashed line), observation variance in observation subspace. Exe-E and Obs-E projections correspond to those shown in (**A**), Obs-O projection corresponds to the denominator of alignment measure. *Right Panel:* Alignment index of observation activity in the movement subspace (purple horizontal line). Execution alignment index is equal to one by definition (not shown). Scattered grey points show alignment values from the null distribution, and p-values denote proportion of alignment values in null distribution greater than true alignment (**C**) Same as B., but for M1-UIDs. (**D**) Same as B, but for F5. (**E**) Bootstrap distributions of alignment values for WHG projected onto PG-defined axes, for execution and observation in each sub-population. P-values denote proportion of execution alignment values greater than observation values. (**F**) Same as (**E**), but for PG projected onto WHG axes.

The online version of this article includes the following figure supplement(s) for figure 7:

**Figure supplement 1.** WHG execution and observation activity in execution movement subspace.

and tested the decoder with observation and NoGo conditions only (*Figure 8C*). F5 showed a significant decoding between these two conditions 150 ms after the imperative cue, whereas for M1-UIDs and M1-PTNs, this was delayed until 300 ms, suggesting that observation and NoGo shared a more similar initial profile in M1 populations. We also trained and tested the decoder on the other condition pairs (Execution-Observation, Observation-NoGo), and these also always produced strong decoding from 100 to 150 ms after the Go/NoGo cue. To examine this further, we performed a second PCA (*Figure 9* and *Figure 9—figure supplement 1*) this time defining each population's subspace using observation activity after the Go cue (see Materials and methods). We then projected each condition's activity onto this subspace, which allowed us to compare the overlap of the execution and NoGo conditions with the observation subspace separately, in an analogous way to the analysis presented in *Figure 7*. In M1-PTN and M1-UID populations, NoGo trajectories (orange) show a closer similarity to observation ones (purple) (*Figure 9A* and *Figure 9—figure supplement 1A*, left and middle panels). Although the M1-PTN population trajectory during NoGo condition showed smaller variance, its evolution over time was similar to the observation population trajectory, with the 'trough' of both trajectories occurring at a similar time in advance of the average time of experimenter HPR (orange filled circles). By contrast, execution activity (green) showed quite different patterns to observation. In F5 (right panel), the execution and NoGo trajectories both showed little variance, suggesting that neither condition overlap strongly with the observation subspace. Quantitatively (*Figure 9B* and *Figure 9—figure supplement 1B*), M1-PTN NoGo activity overlapped

with observation activity during this period significantly more often than chance (PG alignment: p=0.0001, WHG: p=0.0001), and the raw alignment value was much larger for NoGo than for execution (PG NoGo: 0.32, execution: 0.05; WHG NoGo: 0.39, execution: 0.19). M1-UID NoGo activity also overlapped significantly with observation relative to chance (PG: 0.11, p=0.0007, *Figure 9C*; WHG: 0.26, p=0.0001, *Figure 9—figure supplement 1C*), whereas execution activity did not (PG: 0.01, p=0.67; WHG: 0.03, p=0.80). F5 NoGo and execution activity showed low levels of overlap with observation during this period, although this was significant for WHG (PG NoGo: 0.03, p=0.26, execution: 0.02, p=0.47, *Figure 9D*; WHG NoGo: 0.11 p=0.0011, execution: 0.09, p=0.22, *Figure 9—figure supplement 1D*). A split-trial analysis based on EMG magnitudes in the NoGo condition did not affect any of the results, likely because deviations from baseline EMG during NoGo sufficient for trials to be discarded were even rarer than those during observation.

## Discussion

Early work on motor area responses during action observation presupposed that this activity did not result in overt movement in the observer because it was largely absent in M1, and especially within the direct corticospinal projections critical to skilled movement. Although evidence against this hypothesis came from the finding that many PTNs in F5 and M1 can be active during action observation (*Kraskov et al., 2009*; *Vigneswaran et al., 2013*), reduced activity in some M1 neurons during action observation still conformed to a threshold-based explanation for how movement is withheld in this condition. In this study, we considered whether the temporal pattern of F5 and M1 population activity during the execution and observation of naturalistic grasping could provide a state-based explanation as to how observation activity is prevented from resulting in inadvertent movement. We

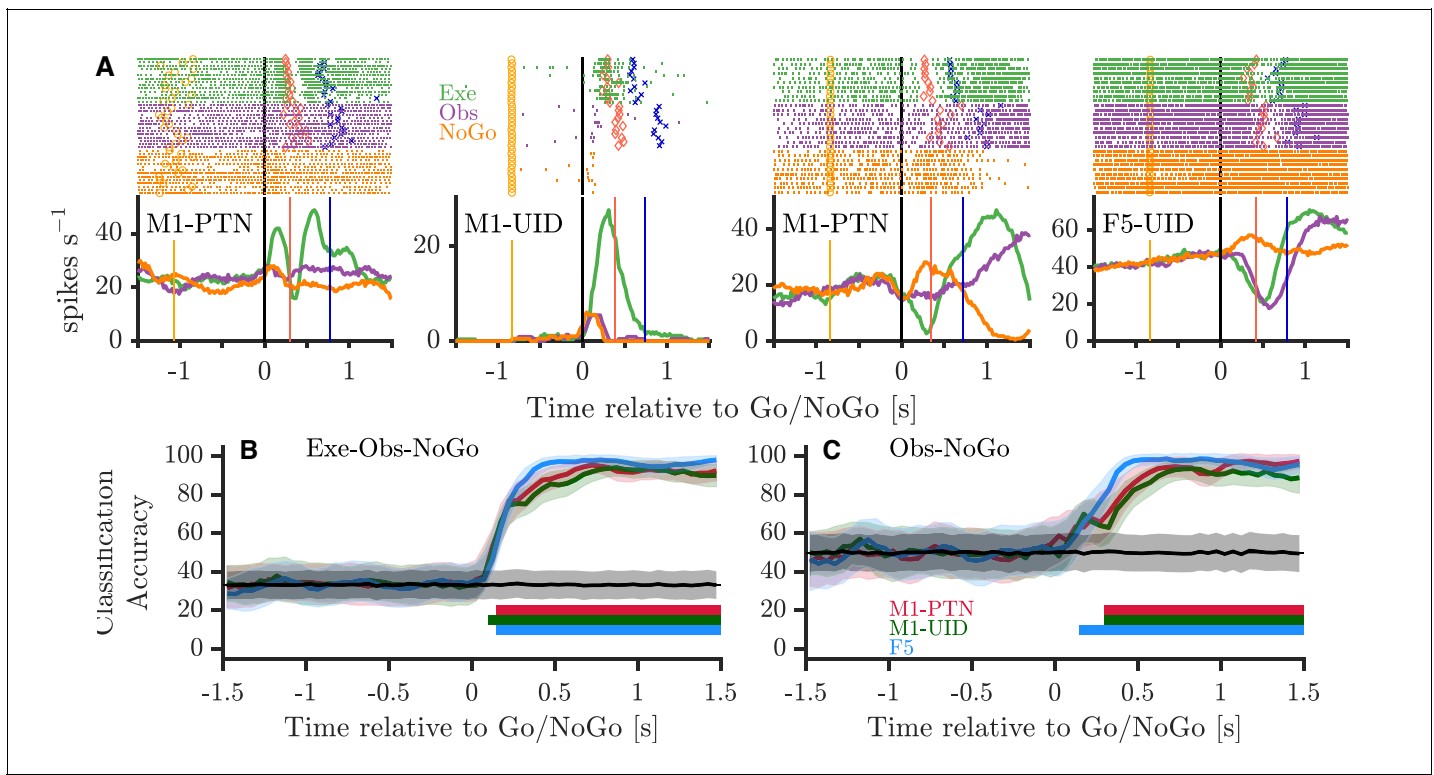

**Figure 8.** Activity during NoGo. (A) Example single-neuron responses during execution, observation, and NoGo. Each subplot shows a raster and histogram representation of single-neuron activity during PG execution (green), observation (purple), and NoGo (orange), with single alignment to the Go/NoGo cue (vertical black lines). Rasters and histograms are compiled from a randomly selected subset of 10 trials in each condition. For histograms, firing rates were computed in 20 ms bins and boxcar-smoothed with a 200 ms moving average. Event markers colour-coded as shown previously (*Figure 1C*). (B) Classification accuracy of maximum correlation coefficient classifier decoding between execution, observation, and NoGo conditions within each population. Grey trace and shading shows mean ±1 SD of decoding accuracy following permutation shuffling, and coloured bars along bottom show period of consistent significant decoding for each population. (C) As for (B) but decoding between observation and NoGo only.

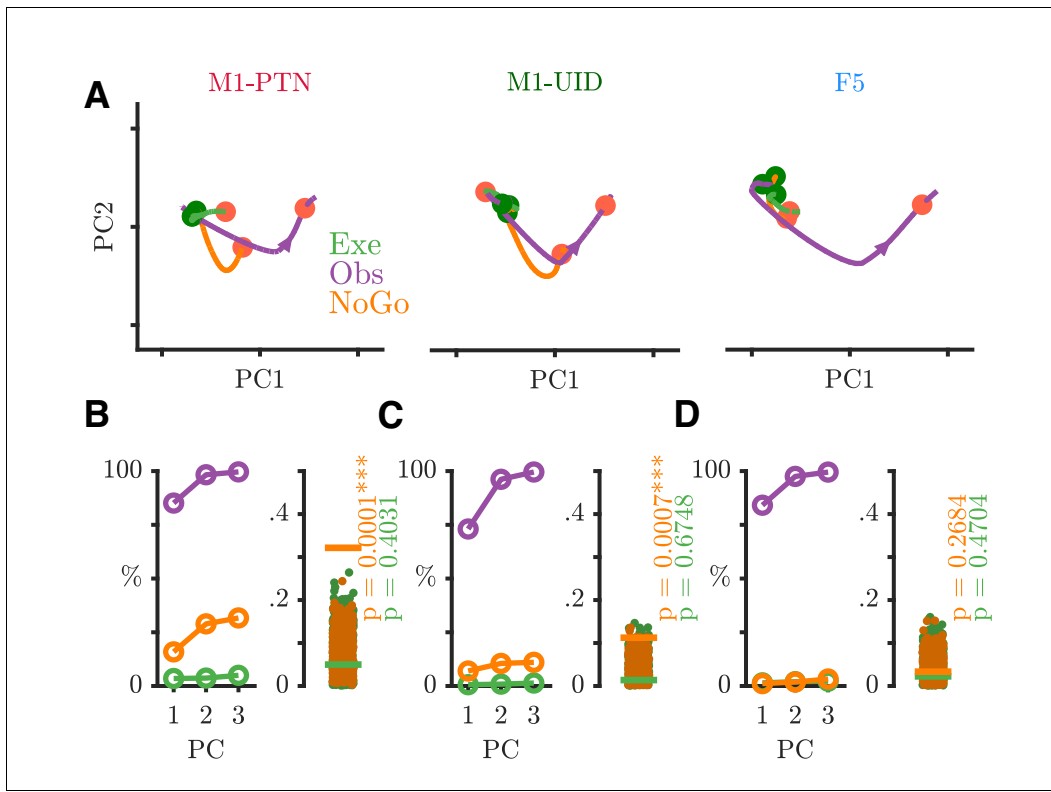

**Figure 9.** NoGo activity within an observation subspace. (**A**) Traces showing the evolution of M1-PTNs, M1-UIDs and F5 population activity during PG execution (green), observation (purple) and NoGo (orange) conditions within the first 2 dimensions of an observation subspace spanning the 100–400 ms after the Go cue. Each trajectory shows the −100 to +400 ms period around the Go/NoGo cue (green/red circles). Average HPR time (across execution and observation) is also indicated on each trajectory by the orange filled circles. The purple arrow on observation trajectories indicates the direction of time. (**B**) M1-PTNs *Left Panel:* Cumulative variance captured within the first three principal axes for execution, observation, and NoGo. *Right Panel:* Alignment indices of execution and NoGo activity in the observation subspace shown as coloured lines (Execution - green, NoGo - orange). Observation alignment index is equal to one by definition (not shown). Scattered points show alignment values from null distributions for execution and NoGo separately, and p-values denote proportion of alignment values in null distribution greater than alignment in data. (**C**) Same as B., but for M1-UIDs. (**D**) Same as B, but for F5.

The online version of this article includes the following figure supplement(s) for figure 9:

**Figure supplement 1.** WHG observation and withholding.

first found that both the modulation depth and profile of activity in F5 MNs was more similar between execution and observation. In M1 populations, particularly M1-PTNs, although many neurons did modulate during both execution and observation, both the magnitude and pattern of activity was distinct between the two conditions. Furthermore, initial observation activity in M1 overlapped with activity when the monkeys simply withheld their own movement, suggesting that action observation can elicit movement suppression by evolving through a 'withholding' subspace.

## Anatomical constraints on the outflow of cortical mirror activity

Previous useful interpretation of mirror activity has almost always been made in the context of known motor properties of the areas and pathways in question. F5 is critical for goal-directed visual guidance of the hand (*Godschalk et al., 1981*; *Weinrich and Wise, 1982*; *Rizzolatti et al., 1998*; *Fogassi et al., 2001*), and contains a vocabulary of motor acts (*Rizzolatti et al., 1988*), supporting internal representation of different grasps (*Murata et al., 1997*; *Raos et al., 2006*; *Umilta et al., 2007*; *Spinks et al., 2008*; *Fluet et al., 2010*; *Schaffelhofer and Scherberger, 2016*). F5 makes only a limited contribution to the CST (*Dum and Strick, 1991*; *He et al., 1993*), but is anatomically

(*Muakkassa and Strick, 1979*; *Godschalk et al., 1984*; *Matelli et al., 1986*; *Dum and Strick, 2005*), and functionally (*Cerri et al., 2003*; *Shimazu et al., 2004*; *Schmidlin et al., 2008*; *Kraskov et al., 2011*) strongly interconnected with M1. M1 provides the major drive to the CST and exerts a direct influence over distal hand musculature, which is probably exploited by executive commands necessary for control of skilled hand movements (*Kakei et al., 1999*; *Brochier et al., 2004*; *Lemon, 2008*). In a classical gating model of corticospinal control where increased activity in excitatory pyramidal cells drives movement, the net disfacilitation of M1-PTNs during observation provides a plausible substrate for inhibiting movement, given their anatomical and functional proximity to the spinal output (*Kraskov et al., 2009*; *Vigneswaran et al., 2013*). However, suppression of PTN activity has also been reported during movement execution tasks (*Kraskov et al., 2009*; *Quallo et al., 2012*; *Vigneswaran et al., 2013*; *Soteropoulos, 2018*), and was observed in the present task (*Figure 4C* and *Figure 5A–C*). PTN suppression during movement could drive downstream inhibitory spinal circuits, given that PTNs not only make direct connections with motoneurons via the cortico-motoneuronal (CM)system (*Lemon, 2008*; *Rathelot and Strick, 2009*), but also connect to segmental interneurons within the spinal cord (*Kuypers, 1981*), and tightly timed suppression of muscle activity is essential for skilled movement (*Brochier et al., 2004*; *Quallo et al., 2012*). An alternative, but not mutually exclusive, possibility, is that population activity at the cortical level evolves within a dynamical system, which can implicitly gate downstream circuitry (*Kaufman et al., 2013*; *Elsayed et al., 2016*). However, this framework has largely considered neurons within a given area, albeit physiologically heterogeneous, to be anatomically homogeneous, and has therefore not yet been reconciled with the known anatomy of neuronal sub-populations. Since M1-PTNs retain a privileged position in volitional control (*Lemon, 2008*), a key aspect of this study involved the consideration of how execution and observation activity evolved in this specific population. We first ruled out the possibility that small changes in EMG in these conditions could account for the modulation patterns, particularly in M1-PTNs, by excluding trials in which EMG was detected. Although monkeys were well trained and such trials were generally rare, they were occasionally present, underscoring the importance of simultaneous EMG recordings to verify that M1-PTN activity during observation reflects a true mirror response.

## M1 observation activity is dissimilar to execution activity

We first confirmed that, although both F5 and M1 neurons can show mirror responses (*Figure 4*), F5 mirror activity during observation is more comparable in amplitude to execution activity (*Figure 5*). This is in line with previous reports of F5 MN activity, suggesting a similar representation of grasp irrespective of whether the action is executed or observed (*Gallese et al., 1996*; *Kraskov et al., 2009*; *Bonini et al., 2010*). By contrast, M1 was first thought to completely lack MNs (*Gallese et al., 1996*; *Nelissen et al., 2005*), and although several studies have now shown that neurons in this area, including PTNs, can show mirror responses, this activity is often relatively weak (*Dushanova and Donoghue, 2010*; *Vigneswaran et al., 2013*). Here, we found that M1-PTNs which increased firing during both execution and observation (facilitation-facilitation, or classical MNs), showed a 3- to 4-fold reduction in activity during observation relative to execution (*Figure 5*), quantitatively comparable to previous reports (*Dushanova and Donoghue, 2010*; *Vigneswaran et al., 2013*). Furthermore, M1-PTN MNs also showed a particularly weak correlation between the two conditions during the early stage of movement (*Figure 6A*), and low-dimensional subspaces capturing variance associated with movement execution captured meaningful observation variance in F5, but not in M1-UID and M1-PTN populations (*Figure 7*). Interestingly, PG M1-PTN alignment increased moderately when calculated using the trials with slightly higher observation EMG levels compared to those with lower EMG. Although it is unsurprising that this change was subtle, since both EMG levels were close to baseline EMG and larger EMG changes on these trials would likely have produced errors due to inappropriate homepad release, this result supports the concern that small EMG increases during observation can contaminate neural recordings and potentially introduce spurious 'mirror' effects. During the movement period, F5 grasp subspaces also captured similar levels of variance related to the other grasp during observation and execution, whereas M1 populations captured significantly less 'other grasp' variance during execution. Although direct quantitative comparisons across populations are difficult to interpret as the total dimensionality (i.e. number of neurons) influences the raw alignment value, the similar alignment values during execution agree with previous evidence that the magnitude of selectivity for different objects during the grasp period

is similar in F5 and M1 (but is earlier in onset and more persistent in F5) (*Umilta et al., 2007*). During grasping observation on the other hand, the level of selectivity is similar in F5, but markedly reduced in M1, as shown in the significantly higher overlap between the two grasps during observation. The finding that the patterns of execution and observation activity are more similar in F5 than in M1 is also consistent with recent work demonstrating MN activity in premotor cortex (PMv) and M1 during execution of reach and grasp to be associated with a series of hidden states, which were recapitulated during observation in PMv, but not M1 (*Mazurek et al., 2018*). Since the balance of excitation and inhibition at the motor cortical level are fundamental for movement generation and suppression, then it should be expected that the respective patterns of activity during execution and observation will be reflected in the resultant behaviour. In line with this, the present results indicate that M1 activity during execution and observation, particularly in PTNs, may be sufficiently dissimilar so as to ensure movement is only produced in the former condition. We note that differences between PTNs and UIDs in M1 were not always clear, likely because the UID population reflects a mixed population of interneurons and pyramidal cells (*Soteropoulos, 2018*), including some possibly unidentified (e.g. high-threshold) PTNs. Although classification of putative PTNs from an unidentified population of neurons has been suggested based on spike width, this classification is unreliable in non-human primates (*Vigneswaran et al., 2011*).

The timing and kinematics of monkey and experimenter movements were clearly different, which could explain why similarity between execution and observation decreased during the reaching phase, however, there are several reasons this is unlikely to be a dominant factor. Firstly, correlations between execution and observation already began to decrease during the late reaction period, that is, before any movement had occurred (*Figure 6A*). At the single-neuron level, firing rates showed little correlation with movement speed (inversely proportional to movement time given constant distance between hand and objects) (see also *Vigneswaran et al., 2013*). Furthermore, given that many sessions involved simultaneous recording of units in F5 and M1, timing reasons could not explain differences between the sub-populations. The targeting of recordings to F5, an area with a preponderance of grasp-related activity (*Rizzolatti et al., 1988*; *Gallese et al., 1996*; *Raos et al., 2006*; *Umilta et al., 2007*; *Michaels et al., 2018*), and the M1 hand area, may also contribute to closer similarity between execution and observation during grasp and hold, rather than reach periods of the task. However, we did not impose strong online selection criteria regarding the proximal vs. distal related activity of recorded cells (in particular, all stable and well-isolated PTNs, once identified, were recorded for a full set of trials), and although our recordings were restricted to M1 and the area of premotor cortex inferior to the arcuate spur (*Figure 3*), rICMS at some recording sites elicited movements of proximal muscles. This is also consistent with a developing body of literature involving anatomical tracing, stimulation mapping and assessments of task-related activity which questions the simple segregation of dorsal and ventral premotor cortex into reaching and grasping areas, respectively (*Raos et al., 2003*; *Dum and Strick, 2005*; *Stark et al., 2007*; *Lehmann and Scherberger, 2013*; *Takahashi et al., 2017*). Nonetheless, there is now ample evidence that cells in dorsal premotor areas, or within proximal limb representations in M1, do mirror reaching movements (*Cisek and Kalaska, 2004*; *Dushanova and Donoghue, 2010*; *Papadourakis and Raos, 2019*), and are key to the generation of reaching actions (*Tanji and Evarts, 1976*; *Churchland and Shenoy, 2007*; *Churchland et al., 2012*). To our knowledge, the anatomical identity of these MNs, and their potential influence on downstream targets, has not been directly tested, but would likely be of particular relevance for initiation or suppression of reaching movements.

## Movement suppression in the lead up to action observation

The dissociation between execution and observation appeared most prominent in the lead up to movement onset, in line with previous suggestions regarding the role of MNs in movement suppression (*Kraskov et al., 2009*; *Vigneswaran et al., 2013*). This presents the possibility that movement is withheld during observation simply by virtue of a withdrawal of sufficient excitatory drive within spinal outputs, or that active suppression processes are involved. These two processes could and probably do coexist, as suggested by the simultaneous presence of classical mirror neurons with weak facilitation responses, and suppression mirror neurons which reduce firing below baseline during observation. To examine whether population-level observation activity might reflect an active, general mechanism for movement suppression, we considered whether observation activity aligned with activity during another simple form of movement suppression, the NoGo condition. We

identified movement-related cortical neurons responding to both observation and NoGo conditions to varying degrees (*Figure 8A*). A decoder trained to discriminate between three conditions exceeded chance and reached plateau 100–150 ms after the Go/NoGo cue (*Figure 8B*), presumably the time necessary for visual information about trial type to become available to motor areas. A second decoder trained to distinguish only between observation and NoGo took longer to exceed chance performance for M1 populations, indicative of similar activity patterns in the two conditions (*Figure 8C*). This was corroborated by analysis of the evolution of activity within an observation subspace after the Go cue, which captured significant NoGo variance in M1-PTNs, but less so in F5 (*Figure 9*). Taken together, these results demonstrate a greater overlap between observation and NoGo neural states in M1 than F5, and support the suggestion that passive action observation triggers a general mechanism for the withdrawal of descending drive from M1 and the subsequent inhibition of unwanted self-movement.

We consider several aspects of the task design particularly relevant to our results, including the use of a pseudo-randomised trial sequence, and the fact that Go/NoGo and execution/observation information was provided at the same moment on each trial (Go/NoGo cue; *Figure 1B*). This meant that the timing of the salient cue to generate or refrain from movement was equivalent across conditions, and monkeys could not anticipate the trial type ahead of this time through any alternative cues. This set-up contrasts with most action observation studies in which block-designs are used, and provides a more ethologically valid framework for assessing functions of the CST in movement suppression, since real-world action execution and observation often take place in quick succession, and appropriately timed generation or suppression of movement is therefore critical to behaviour. The fact that the objects were within the monkey's reach was in part determined by the requirement that trial cues were ambiguous until the Go/NoGo cue, but may also have influenced our findings. Observed actions occurring in peri-personal space often modulate MN responses differently to when the action is beyond the monkey's reach (*Caggiano et al., 2009*; *Bonini et al., 2014a*; *Maranesi et al., 2017*), suggesting the capability to interact with observed actions is a contributing factor to mirror activity. Alternative task set-ups which provide different contexts, such as block designs or those in which observation takes place in extra-personal space, would likely alter the relationship between action observation and action suppression dynamics.

At least in the current task, the difference between F5 and M1 is critical, as it suggests that while M1's priority is to distinguish movement from non-movement from an egocentric perspective, F5 maintains a more similar representation across executed and observed actions, independent of the acting agent's identity. These results suggest the formulation of a simple model framework, in which the movement execution and suppression features of the unfolding action observation response in M1 (and F5) reflect a balance of the activity patterns seen during the execution and NoGo conditions. This balance could be determined by inputs from upstream areas within the MN system, and prefrontal areas responsible for encoding general features of action and self versus other encoding in different contexts, as well as intrinsic dynamics within premotor and motor cortex. State-space analyses, such as those used here, provide a useful tool for analysing these temporal dynamics during different stages of action execution, observation, and withholding. Several avenues for future investigation would likely provide further insights into the evolving dynamics of action execution and observation activity. A wider sampling of grasping execution state space (i.e. recording from more neurons and doing so simultaneously, but also using a much more extensive range of movement and grasping conditions, within a well-defined hierarchical structure) would enable a more detailed assessment of the similarity of action representation across the execution and observation of different grasping behaviours. The increasing possibilities for simultaneous recordings of a larger number of neurons hold particular promise for exploring the trial-to-trial process of appropriate action selection within an execution-observation paradigm, although our dataset, with small samples of simultaneously recorded cells per session, was not well suited to this type of analysis. Single-trial analyses may be particularly interesting in conjunction with analysis of eye movements, which have previously been demonstrated to modulate the firing of at least some MNs (*Maranesi et al., 2013*). Since the monkeys in our task were able to gaze freely, it is possible that observation trials in which the grasp was actively attended would show greater similarity to execution than trials in which gaze was averted. Causal perturbation experiments in conjunction with state-space analyses could provide supporting evidence that action observation activity partly evolves within a 'withholding' subspace, if for example, thresholds for inducing movement during observation were dependent on stimulation

time, or observing congruent or incongruent actions differentially affect action execution. This withholding subspace could also be characterised further using, for example, a stop-signal reaction time (SSRT) task (*Pani et al., 2019*) where failed-stop trials are frequent, although the implementation of this within an action observation paradigm is not straightforward and requires careful consideration.

## Conclusions

In this study, we confirm that F5 activity is closer in amplitude and profile during action execution and observation, whereas there is a particularly weak temporal relationship in activity between the two conditions in M1 populations, including within an identified group of PTNs. The M1 neural state during observation diverges from the execution state in the lead-up to movement onset, and instead appears closer to an action withholding state at this time. Functionally, the different patterns of activity between execution and observation in the two areas could support a context-dependent dissociation between grasp-related visuomotor transformations and the recruitment of descending pathways for elaboration into actual performance of skilled grasp. The increasing capabilities for wide-scale simultaneous recordings from many neurons, identification of neuron subtypes, and accompanying inactivation and manipulation experiments, should help to shed further light on the transfer of information through defined premotor and motor populations for the representation and organisation of goal-directed actions, and the observation of these actions.

# Materials and methods

## Monkeys

Experiments involved two adult male purpose-bred rhesus macaque monkeys (*Macaca mulatta*, M48 and M49, weighing 12.0 kg and 10.5 kg, respectively). All procedures were approved by the Animal Welfare and Ethical Review Body at the UCL Queen Square Institute of Neurology, and carried out in accordance with the UK Animals (Scientific Procedures) Act, under appropriate personal and project licences issued by the UK Home Office. The monkeys were single-housed based on veterinary advice, in a unit with other rhesus monkeys, with natural light and access to an exercise pen and forage area. Both monkeys gained weight regularly throughout the procedure. At the end of all experiments, both monkeys were deeply anaesthetised with an overdose of pentobarbital and perfused transcardially.

## Experimental task

In each session, the monkey sat opposite a human experimenter, with a custom-built experimental box apparatus between them (*Figure 1A*). The monkey was presented with two target objects in peri-personal space, a trapezoid affording PG, and a sphere affording WHG (*Figure 1A*, inset). Each trial began after a short inter-trial interval (ITI) (1–2 s), with the monkey depressing two homepads with both hands and the experimenter depressing a homepad on their side. A controllable LCD screen (14 cm x 10 cm) became transparent (LCDon, *Figure 1B,C*), and the object area was illuminated with white light. After a delay (0.25 s in M48, variable 0.25–0.45 s in M49), two amber LEDs illuminated on one side or the other to indicate the target object for the current trial (ObjCue). After a further delay (0.8 s in M48, variable 0.8–1.2 s in M49), a single green or red LED indicated the trial type. When a green LED was presented on the monkey side (Go), the monkey released the active (right) homepad (HPR), and made a reach-to-grasp movement towards the target object using their right hand. The monkey then grasped the object using a trained grasp (DO), rotated the object into a window (>30 rotation) and held for 1 s (HO to hold off). A constant frequency tone indicated that the monkey was in the hold window, and a second, higher frequency tone after 1 s indicated successful completion of the hold. The monkey then released the object and returned to the homepad, and another high frequency tone indicated correct completion of the trial. The experimenter remained still, with their homepad depressed for the duration of the trial. Observation trials followed the same sequence with roles reversed, such that the experimenter performed the same reach-to-grasp and hold movement in front of the monkey, who remained still, with both hands on the homepads. On NoGo trials, a red LED required the monkey (and experimenter) to simply remain on the homepads for the duration of the trial. After a delay (0.7 s in M48, 1.0 s in M49), a single tone indicated the end of the trial. The monkey was manually provided with a small fruit reward directly to

the mouth by the same experimenter following each successfully completed execution, observation or NoGo trial. Fruit rewards were randomly varied in type across trials, although the proportion of higher-valued rewards was increased in the latter stages of some recording sessions to maintain motivation. All trial types were presented in pseudo-randomised order, with relative proportions of 8:3:2 for each object. The larger proportion of execution trials were used to ensure the monkeys remained attentive and were regularly preparing to move. Error trials, where there was a failure to respond appropriately within the constraints of the task (e.g. releasing the homepad before the Go cue), triggered a low frequency error tone and were immediately aborted by the experimental software. The monkey was not rewarded and these trials were excluded from further analysis.

## Surgical implants

To prepare for recordings, subjects underwent several, well-spaced, surgical procedures under full general anaesthesia (induced with ketamine i/m 10 mg/kg, maintained on 1.5–2.5% isoflurane in oxygen). First, a custom-designed TekaPEEK headpiece was secured to the skull for stable head fixation. In further surgeries, after the animal was fully trained, a) a TekaPEEK recording chamber was fixed with dental acrylic and bone cement to cover a craniotomy extending over primary and ventral premotor cortex; b) two tungsten stimulating electrodes were stereotaxically implanted in the left medullary pyramid c) subcutaneous recording electrodes were chronically implanted in up to 12 arm and hand muscles for EMG recording. After each procedure, animals were recovered overnight in a padded recovery cage, and received post-operative analgesic and antibiotics as prescribed under veterinary advice.

## Neuronal recordings

We used 16 and 7 channel Thomas Recording drives (Thomas Recording GmbH, Geissen, Germany), each containing 1–5 quartz glass-insulated platinum-iridium electrodes (shank diameter 80 μm, impedance 1–2 MΩ at 1 kHz) to record in the arm/hand regions of M1 and F5. On a given recording day, we either carried out dual recordings, recording in M1 using the 16-drive, and in F5 using the 7-drive, or recordings in one area using a single drive. Linear array heads (spacing between adjacent guide tubes = 500 μm) were used for initial mapping of M1 and F5, and subsequent recordings were conducted with square (16 drive) or circular (seven drive) heads to target more specific locations (305 μm spacing). Penetration coordinates were estimated using a custom mapping procedure, based on triangulation of chamber lid coordinates measured in drive co-ordinates to an orthogonal system defined by stereotaxic coordinates of the same points measured during implantation of the recording chamber. Penetration locations and orientations (*Figure 3*) were estimated via a geometrical transformation between recording drive and MRI coordinates. Penetrations were made in the left (contralateral) hemisphere of each monkey, and aimed at the inferior bank of the arcuate sulcus (F5), and the hand/arm area of M1, just anterior to the central sulcus. Electrodes were independently lowered using custom computer software and adjusted in depth to isolate single unit activity as clearly as possible (*Baker et al., 1999*). Broadband signals from each drive were pre-amplified (x20, headstage amplifier), further amplified (x150), bandpass-filtered (1.5 Hz–10 kHz), and sampled at 25 kHz via a PCI-6071E, National Instruments card. We simultaneously recorded electromyographic activity from up to 12 muscles in the contralateral arm and hand, and analog signals of object displacement and homepad pressure (5 kHz), as well as the precise timing of all task events at 25 kHz resolution. All data were stored on laboratory computers for offline analysis. After recording at a site, rICMS was delivered via an isolated stimulator. Sequences of 13 pulses at 333 Hz (duty cycle 0.5 Hz) were delivered every 1–1.5 s at intensities up to 30 μA (M1), or 60 μA (F5).

## PTN identification

While searching for cells, pyramidal tract (PT) stimulation was delivered between the two PT electrodes. The search stimulus intensity was 250–350 μA, and pulses were delivered every 0.6 s (biphasic pulse, each phase 0.2 ms). PTNs were identified as well-isolated cells which showed a robust and latency-invariant response (jitter ≤ 0.1 ms) to PT stimulation. Double pulse search stimuli (separated by 10 ms) were used to further help distinguish antidromic v.s. synaptic responses (*Swadlow et al., 1978*). We recorded the antidromic latency of each PTN, determined threshold, and used discriminated spontaneous spikes to collide the antidromic response, providing unequivocal identification of

a PTN. PTN identification was always performed before task recordings, so this sample of cells was unbiased in terms of task-related activity.

## Spike discrimination

Offline spike sorting was performed using modified WaveClus software (*Quiroga et al., 2004*; *Kraskov et al., 2009*). Broadband data were first high-pass filtered (acausal 4th order elliptic 300 Hz-3kHz, or subtraction of a median-filtered version of the signal). Threshold crossings were then sorted into clusters using an extended set of features, including wavelet coefficients, amplitude features, and the first three principal components. PTN spike shapes during task recordings were compared to the recorded waveforms of spontaneous spikes which resulted in successful collisions (*Lemon, 1984*; *Kraskov et al., 2009*). Single units were considered as those with a clean, consistent waveform and with inter-spike interval histograms uncontaminated below 1 ms for bursting units.

## Data analysis

### EMG and behavioural analysis

For visualization purposes, EMG data for each channel were high-pass filtered (30 Hz, 2nd order Butterworth), rectified, low-pass filtered (500 Hz, 2nd order Butterworth), downsampled to 500 Hz, and smoothed with a 100 ms moving average. Signals were then aligned to the Go cue on individual trials, normalized to the 99th percentile amplitude across all trials and then averaged across trials within each condition. We recorded the timing of all relevant task events for subsequent alignment to analog signals. We defined reaction time on each execution and observation trial as the time between the GO cue and HPR, and movement time as the time between HPR and DO. For visualization of displacement and homepad signals (*Figure 2* and *Figure 2—figure supplement 1*), individual trials were aligned to the Go cue. Signals were normalized to the 99th percentile amplitude across all trials and then averaged across trials within each condition.

To quantitatively assess the level of simultaneously recorded EMG activity during different stages of the task, we calculated the mean rectified EMG envelope (0.5–30 Hz, 4th order Butterworth) in different task intervals for each recording session, muscle, and task condition. Noisy channels were defined as those in which execution EMG during the reaching period did not exceed EMG during baseline, and were removed from further analysis (8 channels across 2 of 93 recordings). We applied a modified version of a previously used method to iteratively exclude observation and NoGo trials from each session in which small changes in EMG may have contaminated the neural response (*Kraskov et al., 2009*). For observation and NoGo conditions, and each muscle separately, we compared EMG during the baseline epoch (LCDon-ObjCue) to EMG during the Reaction period (Go-HPR for observation, 0–300 ms from NoGo for NoGo condition), via an unbalanced t-test, and removed the observation or NoGo trial with the largest magnitude if the t-test was significant (p<0.05). We repeated this procedure until the test was no longer significant (p>0.05). After this procedure, two neurons with fewer than 10 observation trials per grasp remaining were excluded from the dataset, and a further six neurons with fewer than 7 NoGo trials per grasp were excluded from NoGo analyses. Across 93 recordings, the mean number of observation trials excluded was 1.02, and within 20 sessions in which at least one trial was excluded, the mean was 4.75 trials (median = 1). For NoGo, the mean number of trials excluded was 0.25, and within 10 sessions in which at least one trial excluded, the mean was 2.3 trials (median = 1).

To construct summary plots of EMG activity after the imperative cue in each condition (Reaction interval), we subtracted the mean and divided by the standard deviation of the baseline interval across trials, and for each recording and condition, calculated the median 2-norm of the M-length vector across trials (M = 12 muscles) that is, the Euclidean distance of each trial's EMG during the baseline and Reaction intervals from the average baseline EMG. We compared these distance metrics across the Baseline interval and Observation/NoGo Reaction intervals via paired-test (n = 93 sessions). We note that the median Euclidean distance of the baseline interval to the mean baseline EMG is not zero, but reflects trial-to-trial variability in EMG. We compared the Observation/NoGo Reaction intervals to the Execution Reaction interval in a similar manner.

We also assessed the effect of small changes in EMG in the lead-up to movement generation or suppression on our subspace analyses. To do this, we performed a median-split of all trials (prior to any EMG-based exclusion) for each object according to the magnitude of the 2-norm during

observation or NoGo Reaction intervals, and computed PSTHs separately for trials with relative EMG magnitudes relatively close or far from EMG during execution, before repeating the subspace analyses. For all these analyses, we selected the Reaction interval to facilitate direct comparison across the three conditions, and because this represented the most likely interval in which monkeys, although well-trained, might occasionally initiate inappropriate movements following the imperative cue.

## Single-neuron analyses

To define MNs, we initially assessed task-dependent modulation during execution and observation within three key epochs - (1) LCDon-CUEon (Baseline) (2) HPR-DO (Reach) (3) 0–700 ms from HO (Grasp/Hold). Firing rates during execution and observation separately were subjected to a 2-way ANOVA with factor EPOCH (three levels), and GRASP (PG, WHG), followed by post-hoc comparisons of the two task epochs to baseline for each grasp. Neurons which showed a significant main effect of epoch or significant interaction, and at least one significant post-hoc result, were considered task-modulated, and neurons modulated during both execution and observation were classified as MNs. We further categorised MNs according to the sign of their maximum modulation during the two task epochs of both execution and observation, for each grasp separately. Thus, MNs could be subdivided into facilitation-facilitation (F-F), facilitation-suppression (F-S), suppression-suppression (S-S), or suppression-facilitation (S-F) types for each grasp, based on their responses to execution and observation, respectively.

## Population analyses

For all population analyses, spike times for each neuron were binned into firing rates, baseline-corrected and normalized, where necessary. The exact details differed for different analyses, and are described in turn below.

### Heatmaps and population averages

To normalize neural population activity during the task, spike counts in 10 ms bins were smoothed with a Gaussian kernel (unit area, standard deviation 50 ms) and converted to spikes $s^{-1}$. As the timing of events varied across trials, conditions and sessions, firing rates were aligned separately to Go, HPR, and DO events on each execution and observation trial as appropriate, so that the relative timing of these three events, covering the most dynamic period of the task, was matched across all conditions and units. For visualisation purposes, PSTHs aligned to different task events were interpolated to produce one continuous firing rate for each condition. The Go/NoGo event was set as time 0, and HPR and DO were defined as the mean times across conditions, objects, and sessions. The average firing rate across conditions in the 250 ms prior to LCDon was subtracted. To prevent high-firing neurons from dominating the analysis, but preserve some relative range of firing rates, we used a previously applied method (*Churchland et al., 2012*) to soft-normalize the resultant net firing rates, by dividing the total firing rate range across all times and conditions, with a small constant of 5 spikes $s^{-1}$ added to the denominator. Each unit's firing rate across all conditions was therefore limited to a maximum theoretical range of $[-1,1]$, where negative normalized values correspond to suppression of the firing rate relative to the baseline (*Kraskov et al., 2009*; *Vigneswaran et al., 2013*).

### Correlation analyses

To make an initial analysis of the correspondence between execution and observation activity across the task, we assessed the correlation between population activity in the two conditions at different timepoints. To do this, we first averaged each neuron's activity separately within eight task periods, and then across trials, for each condition. The eight task periods were as follows: (1) 250 ms period before LCDon (2) Pres: LCDon-CUEon (3) Object Cue: 500 ms period before the Go/NoGo cue. (4) Early React: 0–150 ms from the Go/NoGo cue (4) Late React: 150–300 ms from the Go/NoGo Cue. (6 and 7). Early and Late Reach: the first and second halves of the HPR-DO interval, which varied in length on each trial. (8) Hold: 0–700 ms from HO. The React period was split at 150 ms to reveal differences when visual information regarding the trial type likely became available to the motor system, and similarly, the Reach period was divided into two to provide a finer-grained picture as

dynamics progressed from reaching into hand-shaping and grasp. Activity was baseline-corrected by subtracting the average activity in the 250 ms prior to LCDon, and then soft-normalized by the maximum absolute rate across all epochs and conditions, with a small constant (+5) again added to the denominator to reduce the influence of low-firing neurons and improve interpretability of scatter plots. For each epoch, the net normalized execution and observation activity within a MN population were extracted as a pair of $\mathbb{R}^{NC \times 1}$ vectors (N = number of MNs, C = number of grasps [2]), and the Pearson correlation coefficient between pairs of vectors was calculated. To compare observed correlation values to those expected by chance, we repeatedly shuffled (1000 iterations) the observation vector to destroy any within-unit relationships, and re-calculated the correlation coefficient, generating a null distribution of correlation values. We assessed significance both via the Pearson correlation coefficient p-value, and if observed correlations fell beyond the range of 95% of the values in the null distribution. We observed no qualitative differences when using the Spearman correlation coefficient. To examine the stability of cross-condition similarity in each population, we extended the cross-condition correlation procedure to correlate activity across timepoints, using time-resolved firing rates. To avoid trivial correlations induced by Gaussian smoothed firing rates, we calculated spike rates in 50 ms non-overlapping bins, with the same multiple alignment as used for the population averages (Go, HPR, DO). We then correlated PSTH activity at execution condition timepoint $t$ with activity at all timepoints $t = 1...T$ in the observation condition, and vice versa, and then averaged across the diagonal. This produced a $T \times T$ matrix containing the correlation values of each timepoint $t$ with every other timepoint.

## Decoding analyses

We used the Neural Decoding Toolbox (*Meyers, 2013*) to examine how well activity in each sub-population discriminated between conditions before and after the Go/NoGo cue. We first ran the decoding across all three conditions (Execution, Observation, NoGo), and then repeated the analysis using Observation and NoGo conditions only. Binned data (non-overlapping 50 ms bins), singly aligned to the Go/NoGo cue for each trial, were used to form pseudo-populations of units for each population separately, using 10 trials from each condition (3 × 10 = 30 data points for each condition in the 3-way decoding), and then randomly grouped into 10 cross-validation splits (three data points per split). Firing rates were z-scored to reduce the bias of high-firing units in the classification. A maximum correlation coefficient classifier was trained on all but one of the splits, and then tested on the left-out split, and this procedure was repeated up to the number of splits, leaving out a different split each time. For increased robustness, the cross-validation splits were resampled 50 times, and decoding accuracy was averaged across these runs. To assess the significance of the observed decoding accuracy, we used a permutation test procedure. The classification was performed exactly as for the original data, except the relevant trial condition labels were shuffled beforehand. This was repeated 50 times to generate a null distribution of the decoding expected by chance, and the observed decoding accuracy was considered significant for a given bin if it exceeded all the values in the null distribution. To reduce the false positive rate, bins were considered truly significant only if they fell within a cluster of at least five consecutive significant bins.

## Subspace analyses

To compare the trajectories of MN activity in each sub-population, we applied PCA. PCA identifies an orthogonal transformation for (correlated) data, where each successive dimension in the transformed space captures the maximum possible variance in the data, while remaining orthogonal to all other dimensions. Projection of data onto the leading principal axes can therefore be used to reduce dimensionality in a principled manner, and reveal low-dimensional structure which may otherwise be obscured. To apply this method to our data, PSTHs (firing rates in 10 ms bins, convolved with a Gaussian kernel of unit area and 50 ms standard deviation) were used to form pseudo-population firing rate matrices for each condition and neuronal sub-population. As in previous analyses, firing rates were soft-normalized by the total firing rate range across all times and conditions (+ a small constant of 5 spikes s$^{-1}$). Similar results were obtained with variations of these parameters e.g. 25 ms Gaussian kernel, or alternative choices of soft-normalisation constant (0, +10, +15).

Trial-averaged execution data from 50 ms before the HPR cue to 500 ms after HO, separately for each object, was then used to form a peri-movement activity matrix **M** ($T \times N$, where $T$ was the

number of timepoints and *N* was the number of MNs), which was then centred by subtracting the mean activity across time for each neuron (dimension). We projected trial-averaged execution and observation data spanning this time period onto the first *k* principal axes ($k = 3$; three dimensions typically captured >90% of the variance in **M**), yielding k principal components for each condition, each with a fractional variance associated with it. We quantified the overlap, or 'alignment', of observation activity within this space by normalizing the total captured variance by the maximum variance which could be captured by *k* axes, according to the following equation (*Elsayed et al., 2016*).

$$a = \frac{tr(V_{Exe}^T cov(X_{Obs})V_{Exe})}{tr(V_{Obs}^T cov(X_{Obs})V_{Obs})} \tag{1}$$

$\mathbf{V_{Exe}}$ and $\mathbf{V_{Obs}}$ are the first *k* eigenvectors of $\mathbf{X_{Exe}}$ and $\mathbf{X_{Obs}}$ , where $\mathbf{X_{Exe}}$ and $\mathbf{X_{Obs}}$ are the mean-centred execution and observation activity, respectively. *tr* denotes trace. The denominator is mathematically equivalent to the sum of the eigenvalues of the first *k* eigenvectors of $\mathbf{X_{Obs}}$ and the alignment index is thus bounded between 0 (if $\mathbf{X_{Exe}}$ and $\mathbf{X_{Obs}}$ are fully orthogonal) and 1 (if $\mathbf{X_{Exe}}$ and $\mathbf{X_{Obs}}$ are perfectly overlapping). We compared true alignment values to a null distribution of alignment of 10,000 random, orthonormal subspaces to the execution subspace, and a p-value was computed as the proportion of values in the null distribution greater than the true alignment. $p < 0.05$ was considered significant (i.e. the true alignment value exceeded 95% of the values within the null distribution). We note that the alignment of uniformly random orthonormal subspaces is dependent on the dimensionality, rather than structure, of the data, and therefore constitutes a relatively low bar for significance testing. However, an alternative method which seeks to circumvent this issue by constraining random subspaces to be drawn from the covariance structure of the full dataset (*Elsayed et al., 2016*) is biased towards identifying orthogonality between two different subspaces.

To quantify the similarity between the low-dimensional trajectories for each grasp during execution and observation, respectively, we also calculated the alignment between grasps for each subspace and sub-population. To generate a distribution of alignment values which could be compared between the two conditions, we sub-sampled 50% of the neurons from each population for the PCA and repeated this x1000. Since our a priori hypothesis was that grasps would be more different during execution, we then calculated a p-value as the proportion of bootstrapped execution alignments greater than their corresponding observation alignments.

To assess whether observation activity evolved in a similar subspace to another form of active movement suppression (NoGo), we examined the state-space overlap between observation and NoGo, using PCA to define a second set of 3 principal axes using trial-averaged observation data from across all neurons, 100–400 ms after the Go cue. We then projected activity from all three conditions onto these axes, and quantified variance captured and alignment statistics in an analogous way to that for the movement period subspaces.

## Acknowledgements

SJJ was funded by a Brain Research UK Graduate Student Fellowship. AK was funded by the Wellcome Trust, grant number 102849/Z/13/Z. The authors thank Tabatha Lawton, Dominika Klisko, Adam Keeler, and Yeung-Yeung Leung for help with recordings, and Spencer Neal, Jonathon Henton, Chris Seers, and Martin Lawton for technical assistance. Roger Lemon provided useful feedback on an earlier version of the manuscript.

## Additional information

### Funding

| Funder | Grant reference number | Author |
| --- | --- | --- |
| Wellcome | 102849/Z/13/Z | Alexander Kraskov |
| Brain Research UK | Graduate Student Fellowship | Steven Jack Jerjian |

The funders had no role in study design, data collection and interpretation, or the decision to submit the work for publication.

## Author contributions
Steven Jack Jerjian, Software, Formal analysis, Validation, Investigation, Visualization, Methodology, Writing - original draft, Writing - review and editing; Maneesh Sahani, Supervision; Alexander Kraskov, Conceptualization, Resources, Supervision, Funding acquisition, Project administration, Writing - review and editing

## Author ORCIDs
Maneesh Sahani [iD] http://orcid.org/0000-0001-5560-3341
Alexander Kraskov [iD] https://orcid.org/0000-0002-3576-4719

## Ethics
Animal experimentation: All procedures were designed to minimize discomfort and pain of the animals and were approved by the local Animal Ethics and Welfare Committee and carried out in accordance with the UK Animals (Scientific Procedures) Act (Project Licence 708254). Experiments involved two adult purpose-bred male monkeys (*Macaca mulatta*, M48 and M49, weighing 12.0kg and 10.5kg, respectively). The monkeys were single-housed based on veterinary advice, in a unit with other rhesus monkeys, with natural light and access to an exercise pen and forage area. Both monkeys gained weight regularly throughout the procedure.

## Decision letter and Author response
Decision letter https://doi.org/10.7554/eLife.54139.sa1
Author response https://doi.org/10.7554/eLife.54139.sa2

# Additional files
## Supplementary files
• Transparent reporting form

## Data availability
Matlab codes and data to reproduce Figures 5-7 and Figure 9 are publicly available at https://github.com/sjjerjian/grasp-mirror-neurons (copy archived at https://github.com/elifesciences-publications/grasp-mirror-neurons).

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
