## [Decision Letter]

**Acceptance summary:**

A new paper by Jerjian et al. provides the most thorough investigation of the population activity during grasp execution and observation in primary motor cortex (M1) and premotor cortex (F5) to date. Modulation of activity during action observation was observed in pyramidal tract neurons in both F5 and M1. The neuronal state in M1 during observation, however, was much more similar to a condition in which the monkey needed to withhold a response. This sheds new light onto the way the motor system prevents observation-related activity from causing overt movements.

**Decision letter after peer review:**

Thank you for submitting your article "Movement initiation and grasp representation in premotor and primary motor cortex mirror neurons" for consideration by *eLife*. Your article has been reviewed by three peer reviewers, including Jörn Diedrichsen as the Reviewing Editor and Reviewer #1, and the evaluation has been overseen by Floris de Lange as the Senior Editor. The following individual involved in review of your submission has agreed to reveal their identity: Luca Bonini (Reviewer #2).

The reviewers have discussed the reviews with one another and the Reviewing Editor has drafted this decision to help you prepare a revised submission.

Summary:

This manuscript provides new and interesting evidence about the way in which primates' motor system can code executed and observed actions without invariably turn the latter into an unwanted motor output. Even pyramidal tract neurons can code observed actions, as previously demonstrated by different groups. The population-based analysis of execution, observation, and withholding related activity in M1 and PMd provides new insights into the role of these two areas as crucial nodes in the action observation network. Specifically M1 may dissociate abstract signals about (observed or withheld) grasp representations from executive, descending signals required for actively performing the movement.

Essential revisions:

1) There was some initial uncertainty about what aspect of the paper actually provided a novel insights. The orthogonal subspaces in M1 for execution and observation, and the stronger observation activity in PMd are nice to show empirically, but not necessarily new or unexpected. Ultimately, all reviewers were under the impression that the paper could provide support for the idea that the neural dynamics occurring during withheld actions (in no-go condition) also happen during action observation in M1, but not in F5. This mechanism would make it possible for F5 to represent observed actions with a net facilitation, which is not turned into an unwanted motor output because M1 behaves like during no-go trials. This idea, however, was not very clearly spelled out in the paper and would need much more explaining (see reviewer 1 comment 2, reviewer 2 comment 1, reviewer 3 comments 3-4). Are there alternative interpretations for a similarity of No-go and observation trials in M1? Can the activity be more conclusively shown to be related to withholding?

2) It is not clear why the analysis was split by grasp-type. See reviewer 1 comment 1 and reviewer 2 comment 4. Would it be not informative to analyze whether the observation of an action is similar to the execution of the same action (as compared to another one)? While one may not expect to find a strict visuomotor congruence in the motor and visual selectivity (see Bonini et al., 2014 and Papadourakis and Raos, 2019), it would be important to show how the current results interact with the factor "object"– i.e. comparing the selectivity of grasp representations during execution and observation. At the very least, a combined analysis would hopefully simplify and streamline the result presentation.

3) Finally, many methodological choices (normalization, etc) are not very well justified. The original reviews are appended below and the authors should carefully clarify why the current methods are used.

Reviewer #1:

The paper provides a thorough analysis of mirror-like activity in area F5 and M1 during the execution of two grasp types (precision and whole hand). An additional strength of the paper is the actual identification of M1- PTN neurons. The main conclusions are that F5 shows a correspondence between observation and execution activity, whereas in M1-PTN and M1-UID units, the activity is evolving in relatively independent subspaces.

This alone may explain how the monkey can prevent overt EMG from being generated in the observe conditions. The interpretation of the additional no-go condition in this study is less clear (see point 2).

1) I believe that the paper certainly goes beyond current papers on mirror neuron activity. However, it was not 100% clear what novel conclusions can be really drawn from the paper. Specifically, I was left wondering why the authors conducted the analyses separately for each grasp type. One of the key questions I think is to what degree the difference between neural trajectories that distinguish different types of grasp are re-visited in observation. The current analyses make it impossible to discern whether units that fire more for precision than whole-hand grasp during execution also do so during observations. I think the interesting claim to test is whether observation and execution involve the same action codes. Right now communalities can be driven simply by common variations across the trial time course.

2) The section "movement suppression during action observation" is not clearly motivated. Knowing that there is subspace overlap between No-go and observation activity does per se tell us anything about movement suppression, or does it? These ideas are not spelled out clearer in the Discussion. What are "suppression-like features of the unfolding action observation response in M1"?

Reviewer #2:

This manuscript provides new and interesting evidence about the way in which primates' motor system can code executed and observed actions without invariably turn the latter into an unwanted motor output. Indeed, primary motor neurons can code observed actions, as previously demonstrated by different groups (including the senior author's group), but their activity is remarkably different from that during execution and much more similar to that during a new NoGo condition in which the monkey is required to withhold the action. The findings provide compelling evidence that the primary motor cortex is a crucial node of the action observation network because it may dissociate abstract signals about (observed or withheld) grasp representations from executive, descending signals required for actively performing the movement.

I found the data novel and interesting, and the paper is, in general, technically sound. I have some doubts about data analysis and some suggestions to improve data presentation.

1) During NoGo trials, "a red LED required the monkey to simply remain on the homepads for the duration of the trial". Fine, but please state what the monkey sees exactly in the different conditions, where the experimenter (hand in particular) was in the meantime, and where he/she was during Go condition.

2) I assume the monkey was not fixating: but was there any control of eye position? If not, it should be discussed or stated in some part of the manuscript whether and why possible systematic differences in gaze/attention allocation during the various conditions (e.g. Maranesi et al., 2013) should or should not have influenced the action observation response.

3) Reward. "The monkey received a small fruit reward directly to the mouth for each successfully completed execution, observation or NoGo trials". Who or what (if any automatic device was used) gave the food to the monkey? When was it delivered relative to the tone indicating the end of the trial? From Figure 1C and single neuron examples I assume the reward delivery is outside the illustrated period, but it would be useful to see the response also during this last phase of the task because it is known that the reward can strongly influence the visual response of mirror neurons, at least in PMv (Caggiano et al. 2012 PNAS). In 2 out of 4 single neuron examples (Figure 4A, D) the activity after HPN does not return to baseline level (in Figure 5 the same problem is apparent), supporting the usefulness of seeing what happens next. It should also be stated if the fruit reward was always the same or if it was varied (and, in case, when).

4) I don't understand why some epochs for single-neuron analyses have been (apparently arbitrarily) split into two: this should be at least justified. Furthermore, I cannot understand why baseline has been used in a post-hoc comparison only: it could be reasonable to treat execution and observation separately, given the often radically different magnitude of activity. But why not using a 2-ways ANOVA (rather than one-way) considering EPOCH (including baseline) and OBJECT as factors?

5) It is not entirely clear from Figure 3 if at least some (if not many) of the penetrations were carried out in the ventro-rostral part of the dorsal premotor cortex (which is known to host mirror neurons as well, see Papadourakis and Raos, 2019): showing the reconstruction of the penetration grid on the cortical surface would help.

Reviewer #3:

The submitted paper by Jerjian and colleagues is a detailed investigation of the relationship between single neuron activity in macaque motor cortex (F5 and M1) and the execution, observation and withholding of reaching/grasping actions.

The experimental setup is straightforward. A monkey sits opposite an experimenter and between them are two reachable objects. An amber LED cues whether the animal or person will potentially manipulate the object (depending on whom it is closer to) and which object will be manipulated (depending on whether it is on the left or right side). The amber LED can then turn either green, cueing an execution of the indicated action or red cueing that the indicated action should not be executed. The basic question now becomes how single neurons and the net populations respond to the various parts of this task. Recordings are done with traditional tools over many sessions yielding populations of 100-200 neurons. Of particular note, the authors identify M1 neurons that project to the corticospinal tract, an important feature of the dataset. They also record from muscles to try and control for the animal's state.

Overall, the paper does a careful job cataloging the findings using several approaches. The writing is clear and for the most part very thorough. The results they report are sensible. F5 maintains a more similar representation of action and observation than does M1. M1 is noted to have evidence of a so-called withholding state which may act to dissociate the representation of the grasp from the execution of an action. My general concern relates to how much this really pushes the field forward as well as some methodological issues with respect to the general approach used how they have handled the EMG analysis.

1) What precisely are the authors doing that is new here? Not being directly in this area, it is hard to tell because I do not see a clear statement of the objectives beyond a desire to investigate this interesting topic more carefully than has previously been done. I do see some interesting things like ensuring that trials are randomized order, having the object in peri-personal space, and recording PTNs. Are these things meant to alter our appreciation of this system? If so, how? It comes across that the authors think it’s obvious but without a specific hypothesis being put forth it all falls a bit flat to me.

2) I know that the authors list the limitations of their approach, but it's worth emphasizing how much insight could potentially be gained here by deploying modern approaches on various scales. A simple one would be looking at single trial activity, hardly a big ask these days even in monkeys. A more exciting one would be looking at causal manipulations. ]

3) What precisely do the authors think constitutes evidence for a withholding state? I can't pin them down on it much more than M1 activity goes down. Again, very nicely demonstrated but not definitive (see EMG point below and note that idea of trial-by-trial recordings are missing) and to some degree loose even in the context of the analysis done as far as I can tell.

4) The details about the EMG are absolutely critical here and yet very much lacking. Is EMG analyzed on the same days as the neural activity? Are the analyses performed on a trial by trial basis? Do the experiments have a handle on whether the muscle activity is above threshold during the various epochs of the task? The authors clearly appreciate that one needs to understand the state of the descending motor signals to make the inferences they want to make but then they provide such a high-level overview that it is hard to understand at what level they appreciate the nuances.

5) I don't understand some arbitrary analytical choices. For example, they eliminate neurons that change activity <5 spikes / second (subsection “Single-neuron analyses”). The smoothing window quite broad and acausal (subsection “Heatmaps and population averages”). They add an offset for rate comparisons of +5 for rate condition comparisons (subsection “Correlation analyses”). I appreciate the clarity of the methods, it’s great, but I would like some further details about these kinds of choices and how they influence the results, especially because these are essentially descriptive.

---

## [Author Response]

Essential revisions:1) There was some initial uncertainty about what aspect of the paper actually provided a novel insights. The orthogonal subspaces in M1 for execution and observation, and the stronger observation activity in PMd are nice to show empirically, but not necessarily new or unexpected. Ultimately, all reviewers were under the impression that the paper could provide support for the idea that the neural dynamics occurring during withheld actions (in no-go condition) also happen during action observation in M1, but not in F5. This mechanism would make it possible for F5 to represent observed actions with a net facilitation, which is not turned into an unwanted motor output because M1 behaves like during no-go trials. This idea, however, was not very clearly spelled out in the paper and would need much more explaining (see reviewer 1 comment 2, reviewer 2 comment 1, reviewer 3 comments 3-4). Are there alternative interpretations for a similarity of No-go and observation trials in M1? Can the activity be more conclusively shown to be related to withholding?

In response to fair points raised by the reviewers, we have modified and extended analyses and adapted the text to emphasise the rationale and novel insights of our manuscript. Detailed responses to individual comments are provided below. Here, we highlight the key points.

We believe the present data is more conclusive and thorough in demonstrating the distinction between the pattern of M1 population activity during execution and observation than any previously published data. We provide more definitive evidence that activity in M1 populations, including PTNs, during observation, evolves in movement-null dimensions. Furthermore, the novel comparison to the NoGo condition allowed us to assess whether the observation activity following the Go cue could be reasonably described as related to movement suppression, not just via a dissimilarity to the execution activity, but by a similarity to activity during a canonical form of movement suppression (a NoGo cue). This is not trivial or necessarily expected from the preceding execution-observation analysis, and suggests that a general mechanism for withholding movement is recruited during both observation and NoGo conditions.

One potentially conceivable explanation for the similarity between NoGo and observation conditions in M1 could be that both conditions simply show relatively little modulation, and are therefore inevitably more similar to each other than to execution. On the one hand, it is certainly true that, both in our data and across the literature, activity in these conditions is generally reduced compared to execution. However, we do not believe this can account for the results, which show clear examples of modulation during these conditions in M1 neurons (Figures 4 and 8), temporally specific modulation in population activity (Figures 5 and 6), and time-dependent firing rate variance during observation (Figure 9). In addition, from a mathematical standpoint, if the observation and NoGo data were simply small, random deviations from noise, then subspace overlap would be highly unlikely to exceed chance, since chance itself is defined as the alignment of randomly oriented dimensions to the principal axes. In contrast to this, we showed that in M1 observation and NoGo data overlap significantly more than expected by chance (Figure 9). A second potential explanation could be that the design of the task, with observation and NoGo interleaved, could have introduced some similarity between the conditions. Although we do not think this is sufficient in itself to explain the differences between the F5 and M1 sub-populations, we acknowledge this point in the Discussion and have expanded on the implications of it for the design of action observation tasks.

Perhaps more conclusive evidence to speak to the withholding hypothesis could come from some hybrid observation/SSRT task, although the implementation of such task is not straightforward. We note that causal perturbation experiments (e.g. one might expect to see time-dependent changes in the threshold or probability of inducing movement via stimulation during observation if movement is being actively suppressed), and single-trial analyses would provide further insights into the state-space evolution of activity during the different conditions. Although these were beyond the scope of the study and/or limited in feasibility given the nature of the data, we have expanded the discussion of how these experiments/analyses would be useful.

2) It is not clear why the analysis was split by grasp-type. See reviewer 1 comment 1 and reviewer 2 comment 4. Would it be not informative to analyze whether the observation of an action is similar to the execution of the same action (as compared to another one)? While one may not expect to find a strict visuomotor congruence in the motor and visual selectivity (see Bonini et al., 2014 and Papadourakis and Raos, 2019), it would be important to show how the current results interact with the factor "object" – i.e. comparing the selectivity of grasp representations during execution and observation. At the very least, a combined analysis would hopefully simplify and streamline the result presentation.

The initial rationale for performing analyses on each grasp separately was to focus attention on the differences across execution, observation and NoGo, since we were unconvinced that broad conclusions on grasp-related differences could be made based on two grasps only. However, we accept that across-object similarity of neuronal activity during observation, and how it relates to the differences and similarities in neuronal activity during execution is by itself an interesting question for understanding of mirror neuron properties. We believe that fully addressing such a question can only be achieved with larger and broader set of objects and motor acts employed to interact with the objects, with corresponding hierarchical analyses of object, kinematic, and/or muscle activity relationships (e.g. Raos et al., 2006, Schaffelhofer and Scherberger, 2016). Nevertheless, as suggested here and also by reviewer 2, we now perform a simpler classification of mirror neurons, via 2-way ANOVA (epoch x object). The number of interaction effects is substantially greater during execution, indicative of more frequent object selectivity in this condition, which is consistent with previous literature. In addition, several analyses are now performed across objects, or analyses for both objects are presented in main and supplementary figures. We note that the classification of facilitation-suppression effects is still done within object, since these profiles could be different for different objects.

3) Finally, many methodological choices (normalization, etc) are not very well justified. The original reviews are appended below and the authors should carefully clarify why the current methods are used.

We have streamlined the single-unit classification of MNs (see point 2), and provided additional text justification for several methodological choices such as soft-normalization. We have also confirmed in several cases that although the exact methodological choice of e.g. normalization changes numbers slightly, these changes do not alter our conclusions.

Reviewer #1:The paper provides a thorough analysis of mirror-like activity in area F5 and M1 during the execution of two grasp types (precision and whole hand). An additional strength of the paper is the actual identification of M1- PTN neurons. The main conclusions are that F5 shows a correspondence between observation and execution activity, whereas in M1-PTN and M1-UID units, the activity is evolving in relatively independent subspaces.This alone may explain how the monkey can prevent overt EMG from being generated in the observe conditions. The interpretation of the additional no-go condition in this study is less clear (see point 2).1) I believe that the paper certainly goes beyond current papers on mirror neuron activity. However, it was not 100% clear what novel conclusions can be really drawn from the paper. Specifically, I was left wondering why the authors conducted the analyses separately for each grasp type. One of the key questions I think is to what degree the difference between neural trajectories that distinguish different types of grasp are re-visited in observation. The current analyses make it impossible to discern whether units that fire more for precision than whole-hand grasp during execution also do so during observations. I think the interesting claim to test is whether observation and execution involve the same action codes. Right now communalities can be driven simply by common variations across the trial time course.

We thank the reviewer for their comments. We agree that the distinctions between grasps, and whether these are consistent across execution and observation, provides an interesting test of whether the action codes in the two conditions are similar.

In line with this, and with other reviewer suggestions, we have streamlined classification of mirror neurons and several analyses to include both objects. In our subspace analysis, we now also compare the neural trajectories between the two objects for execution and observation subspaces in our separate populations. This shows that during grasp, F5 trajectories for different grasps show similar degrees of overlap during execution and observation. In M1, execution trajectories are different, whereas observation trajectories are more similar. This is apparent both in the alignment statistics (provided in Figure 7), and correspondingly, profiles of the Euclidean distance between the two objects’ trajectories, which are similar in F5, but less so in M1. Taken together with the results for within-object projections of observation onto execution subspaces, we conclude that relative representation of grasping actions (action codes) is similar across execution and observation in F5, but not in M1.

We agree that commonalities in time course/trial structure between execution and observation could indeed be a concern, and this concern formed part of the rationale for examining overlap only within the reach/grasp period (the , rather than across the whole trial, which inflates alignment statistics substantially given the common structure of the trial, particularly prior to the Go cue.

2) The section "movement suppression during action observation" is not clearly motivated. Knowing that there is subspace overlap between No-go and observation activity does per se tell us anything about movement suppression, or does it? These ideas are not spelled out clearer in the Discussion. What are "suppression-like features of the unfolding action observation response in M1"?

The main motivation for our work was to address an apparent paradox of motor system being active without any production of the movement, which is especially striking because of the anatomical aspect of our research, namely mirror activity within identified pyramidal tract neurons.

In our previous work we suggested that observation activity was related to movement suppression based on an overall dis-facilitation of activity during observation relative to execution, but this still conforms to a simple net firing rate threshold model for movement generation based on a subset of neurons which increased activity during execution. Here we used a population-level approach to evaluate whether the temporal pattern of neuronal activity could explain how observation activity is prevented from resulting in inadvertent movement. In addition, we directly compared observation activity to another form of movement suppression such as NoGo.

Although movement suppression is not probed with causal manipulation, we assume that the NoGo condition elicits a movement withholding state simply by virtue of the task design – the monkey must withhold an anticipated movement toward a particular object, given the matched timing of the execution/observation/NoGo cue on each trial, and the predominance of execution trials, which encourages movement preparation. From this, we assume that if the neural state during observation approaches the same neural state as that during NoGo after the Go/NoGo cue, or vice versa, then both conditions reflect a movement withholding state. As noted above, there are possible alternatives for investigating the ‘withholding subspace’, including use of an SSRT-style paradigm and perturbation experiments, and we have expanded the discussion of these in the paper. We also note that the additional EMG analyses confirm that monkeys are genuinely withholding movement on NoGo (and observation) trials.

Reviewer #2:[…] I found the data novel and interesting, and the paper is, in general, technically sound. I have some doubts about data analysis and some suggestions to improve data presentation.1) During NoGo trials, "a red LED required the monkey to simply remain on the homepads for the duration of the trial". Fine, but please state what the monkey sees exactly in the different conditions, where the experimenter (hand in particular) was in the meantime, and where he/she was during Go condition.

We have added text to clarify that on all trials, the non-acting agent(s) (monkey during observation, experimenter during execution, both monkey and experimenter during NoGo) kept their hand/hands on their homepads (subsection “Experimental task”).

2) I assume the monkey was not fixating: but was there any control of eye position? If not, it should be discussed or stated in some part of the manuscript whether and why possible systematic differences in gaze/attention allocation during the various conditions (e.g. Maranesi et al., 2013) should or should not have influenced the action observation response.

We did not impose any fixation criteria in the task as we believe this reflects a more naturalistic situation for action observation. We have added some discussion of the possible effects of gaze/attention on firing rates, such as those shown in the suggested Maranesi et al. paper (subsection “Movement suppression in the lead up to action observation”).

3) Reward. "The monkey received a small fruit reward directly to the mouth for each successfully completed execution, observation or NoGo trials". Who or what (if any automatic device was used) gave the food to the monkey? When was it delivered relative to the tone indicating the end of the trial? From Figure 1C and single neuron examples I assume the reward delivery is outside the illustrated period, but it would be useful to see the response also during this last phase of the task because it is known that the reward can strongly influence the visual response of mirror neurons, at least in PMv (Caggiano et al. 2012 PNAS). In 2 out of 4 single neuron examples (Figure 4A, D) the activity after HPN does not return to baseline level (in Figure 5 the same problem is apparent), supporting the usefulness of seeing what happens next. It should also be stated if the fruit reward was always the same or if it was varied (and, in case, when).

Small fruit rewards were manually provided by the same experimenter who performed the observation trials. The reward time was thus not locked to the end of the trial, and increases in activity relative to the initial baseline could have arisen due to temporal smearing of motor activity at the time of receiving the reward, or arm movements in anticipation of receiving it after the trial was completed. For this reason, we have not included this period of the task in any of our analyses. The fruit rewards were randomly varied in type, although the proportion of higher-reward fruits was occasionally increased as sessions progressed to maintain motivation. We have added these points to the description of the experimental task (subsection “Experimental task”).

4) I don't understand why some epochs for single-neuron analyses have been (apparently arbitrarily) split into two: this should be at least justified. Furthermore, I cannot understand why baseline has been used in a post-hoc comparison only: it could be reasonable to treat execution and observation separately, given the often radically different magnitude of activity. But why not using a 2-ways ANOVA (rather than one-way) considering EPOCH (including baseline) and OBJECT as factors?

Classification of mirror neurons is now performed via 2-way ANOVA as suggested, and with simplified epochs (Baseline, Reach, Grasp/Hold). We have revised the Materials and methods text to clarify that ANOVAs are conducted separately for execution and observation, followed by comparing each task epoch to baseline. Mirror neurons are only then defined as those neurons modulated in both conditions. We have retained the original epochs for correlation analyses, since this provides a finer-grained picture of the time-varying activity, and demonstrates differences early and late after the Go cue (when visual information likely becomes available to the motor system), and between early and late reach, when predominantly reaching-related activity transitions to hand shaping and grasp.

5) It is not entirely clear from Figure 3 if at least some (if not many) of the penetrations were carried out in the ventro-rostral part of the dorsal premotor cortex (which is known to host mirror neurons as well, see Papadourakis and Raos, 2019): showing the reconstruction of the penetration grid on the cortical surface would help.

We have provided a new version of this figure showing all individual penetrations for both monkeys, confirming that the vast majority of penetrations in which neurons were recorded were inferior to the arcuate spur for F5, and close to the central sulcus and hand knob in M1. From this map, and considering the angle of the penetrations, we consider it highly improbable that any penetrations were carried out in PMd.

Reviewer #3:[…] Overall, the paper does a careful job cataloging the findings using several approaches. The writing is clear and for the most part very thorough. The results they report are sensible. F5 maintains a more similar representation of action and observation than does M1. M1 is noted to have evidence of a so-called withholding state which may act to dissociate the representation of the grasp from the execution of an action. My general concern relates to how much this really pushes the field forward as well as some methodological issues with respect to the general approach used how they have handled the EMG analysis.1) What precisely are the authors doing that is new here? Not being directly in this area, it is hard to tell because I do not see a clear statement of the objectives beyond a desire to investigate this interesting topic more carefully than has previously been done. I do see some interesting things like ensuring that trials are randomized order, having the object in peri-personal space, and recording PTNs. Are these things meant to alter our appreciation of this system? If so, how? It comes across that the authors think it’s obvious but without a specific hypothesis being put forth it all falls a bit flat to me.

We have made revisions to the Introduction, Results and Discussion to more clearly highlight the rationale and motivation for the study and the various analyses, which we accept were not sufficiently well framed. We stress that, despite the importance of our previous work demonstrating relatively reduced activity during action observation in M1 PTNs, the notion that this supports a suppression of movement in this condition was simply a hypothesis, and notably one which still conformed to a simple threshold-based determination of when and whether movement is generated or suppressed, without considering the time-course of activity across the whole population. Although undoubtedly useful, this framework is inconsistent with accumulating evidence indicating that movement generation is driven by the evolution of neural activity through certain points of state space, rather than a simple ramping to threshold across neurons. In light of this, we hypothesised that action observation activity might not produce movement because it resides outside of the movement execution subspace, and that this would be particularly the case for M1-PTNs, but not for example, in upstream F5, where execution and observation activity are generally thought to be more similar (although by no means identical).

Secondly, we hypothesised that if this observation activity instead evolves into a subspace related to suppression, rather than simply an ‘absence’ of movement, it should evolve after the Go cue in a similar manner to activity during another form of movement suppression, which we here consider in the form of a NoGo condition.

We do believe the technical and anatomical considerations of our experiment are important, and have also modified the text to make this more explicit (see subsections “Anatomical constraints on the outflow of cortical mirror activity”, and “Movement suppression in the lead up to action observation”). Firstly, identification of PTNs, given their privileged position in descending control of spinal circuitry, allows for more concrete statements about the potential effects, or absence thereof, of cortical activity on these circuits. Dynamical systems approaches have been leveraged to great effect to make sense of computations within neural populations with heterogeneous response properties, but somewhat paradoxically, often consider these populations to be anatomically homogeneous.

The interleaved trial structure provides a markedly more naturalistic set-up compared to separate execution and observation blocks, which, presumably for technical reasons, are more common. However, a block design would have introduced a number of differences in how monkeys treated the object cue, and subsequent ‘reaction’ period after the Go/NoGo cue, since the trial type would have been apparent from the beginning, and therefore likely removed a trial-by-trial requirement to suppress action on observation trials. Lastly, we speculate that placing the objects in peri-personal space may also increase the salience of action withholding on observation trials, since mirror neurons have previously been suggested to encode operational aspects of the observed action e.g. Caggiano et al., 2009 (‘i.e. can the monkey interact with it?’). It should also be noted that placing the objects in peri-personal space was a technical requirement for unequivocally ensuring that the monkey could not anticipate whether trials would be execution or observation at any time before the Go cue (e.g. based on a different position of the objects).

2) I know that the authors list the limitations of their approach, but it's worth emphasizing how much insight could potentially be gained here by deploying modern approaches on various scales. A simple one would be looking at single trial activity, hardly a big ask these days even in monkeys. A more exciting one would be looking at causal manipulations.

We completely agree with the suggestion that single-trial analyses would provide novel insights regarding the divergence in state-space activity between execution and observation and how action is generated or suppressed. Our recording methods only allowed us to record up to a couple of neurons simultaneously within one area, so the scope for robust single-trial analyses in the present dataset was limited. However, we have revised and expanded the Discussion on the benefits to understanding of execution-observation dynamics which single trial analysis and causal manipulations could provide (e.g. subsection “Movement suppression in the lead up to action observation”).

3) What precisely do the authors think constitutes evidence for a withholding state? I can't pin them down on it much more than M1 activity goes down. Again, very nicely demonstrated but not definitive (see EMG point below and note that idea of trial-by-trial recordings are missing) and to some degree loose even in the context of the analysis done as far as I can tell.

We empirically define a withholding state as the state following the observation of NoGo cue (red LED), which cues the monkey to refrain from producing action (hence the importance of the interleaved trials and ambiguity regarding trial type until the end of the object cue period). We think the reviewer is absolutely right to draw attention to the EMG (also see response to point 4.) – simultaneous recording of EMG alongside neural activity, and the general lack of EMG activity on observation and NoGo trials gives us greater confidence that the monkeys were truly refraining from movements during the period following the imperative cue.

4) The details about the EMG are absolutely critical here and yet very much lacking. Is EMG analyzed on the same days as the neural activity? Are the analyses performed on a trial by trial basis? Do the experiments have a handle on whether the muscle activity is above threshold during the various epochs of the task? The authors clearly appreciate that one needs to understand the state of the descending motor signals to make the inferences they want to make but then they provide such a high-level overview that it is hard to understand at what level they appreciate the nuances.

We agree with the reviewer that EMG details are important for interpreting the relationship between cortical activity and descending drive, and acknowledge that the original submission did not address this satisfactorily. EMG was recorded simultaneously with neural activity, and we have now included several additional analyses to address the reviewer’s concerns. For all neural analyses (except split-trial subspace analyses), we now exclude trials where we detect EMG activity above an empirically defined baseline during observation and NoGo. We also show summary statistics of EMG after the Go/NoGo cue across conditions and recordings (new Figure 2B, C), demonstrating that average EMG in this period only systematically diverges from baseline during execution, as the monkey begins to generate movement. Finally, we briefly show that splitting observation/NoGo trials based on high vs. low EMG has a limited impact on our subspace analyses, although in the case of precision grip for the M1-PTNs, the trajectory produced from trials with overall greater EMG do show a slight increase in overlap with execution. We infer that this increase is small since correctly completed trials with increased EMG during observation are still quite far from execution in ‘EMG space’ (Figure 2C), but given the important role of M1-PTNs in control of voluntary movement, particularly in fine movements such as precision grip, this analysis does suggest that simultaneous EMG recording during action observation is worthwhile to rule out potential spurious mirror effects.

5) I don't understand some arbitrary analytical choices. For example, they eliminate neurons that change activity <5 spikes / second (subsection “Single-neuron analyses”). The smoothing window quite broad and acausal (subsection “Heatmaps and population averages”). They add an offset for rate comparisons of +5 for rate condition comparisons (subsection “Correlation analyses”). I appreciate the clarity of the methods, it’s great, but I would like some further details about these kinds of choices and how they influence the results, especially because these are essentially descriptive.

In line with suggestions by other reviewers, we have simplified some of the analyses (for example the classification of neurons). The smoothing window is acausal for the PSTH analysis (Figure 5) and subspace analyses (Figures 7 and 9), which is common in the literature. However, we have confirmed that results are not impacted by using a narrower smoothing window (25ms Gaussian), and have now indicated this in the Materials and methods (and see Author response image 1). For correlation and decoding analyses, which would certainly be affected by acausal smoothing, we do not perform any smoothing of the data after initial binning of spike counts. We have noted in the methods that the offset of +5 (or similar constant) for soft-normalization of firing rates is a previously used method in the literature to prevent low-firing neurons from having firing rates scaled to a full -1 to 1 range, and thereby contributing excessively to the dimensionality reduction analysis, while also preventing high-firing neurons from dominating the analysis, although the exact choice of constant is not critical to the results.

**Author response image 1. sa2fig1:** Projections of execution and observation activity onto PG execution subspace, in same format as Figure 7A-D, but showing F5-PTNs and F5-UIDs separately. Firing rates were smoothed using a 25ms Gaussian, and normalization constant was set to 0 (i.e. no soft-normalization).